# Breaking the SFT Plateau: Multimodal Structured Reinforcement Learning for Chart-to-Code Generation

**Lei Chen**     **Xuanle Zhao**     **Zhixiong Zeng**[†]
**Jing Huang**     **Liming Zheng**     **Yufeng Zhong**     **Lin Ma**[*]
Meituan
zengzhixiong@meituan.com     forest.linma@gmail.com
Project: https://github.com/DocTron-hub/MSRL

## Abstract

While reinforcement learning (RL) has proven highly effective for general reasoning in vision-language models, its application to tasks requiring deep understanding of information-rich images and structured output generation remains underexplored. Chart-to-code generation exemplifies this challenge, demanding complex reasoning over visual charts to produce structured code. Supervised fine-tuning (SFT) alone is often insufficient, highlighting the need for effective RL strategies tailored to structured outputs. In this paper, we systematically investigate the performance plateau of SFT through large-scale experiments and propose Multimodal Structured Reinforcement Learning (MSRL) for chart-to-code generation. We construct the largest training corpus to date, with 3 million chart-code pairs curated from real-world tables in arXiv papers, addressing the limitations of previous synthetic datasets. Despite achieving state-of-the-art performance, our experiments show that simply increasing SFT data eventually leads to diminishing improvements. To break this plateau, MSRL employs a multi-granularity reward system that integrates both textual and visual feedback. At the textual level, rule-based rewards validate fine-grained code details, while at the visual level, a model-based reward assesses the structural similarity between rendered code and ground-truth charts. We implement a two-stage curriculum training strategy, first optimizing the model with textual rewards and then incorporating visual signals for further enhancement. Experimental results demonstrate that MSRL substantially breaks the SFT plateau, improving high-level metrics by 6.2% and 9.9% on ChartMimic and ReachQA benchmarks, respectively. Notably, our method outperforms all existing approaches in the chart domain and achieves competitive results with advanced closed-source models.

## 1 Introduction

Large language models (LLMs) have demonstrated impressive reasoning capabilities on complex textual problems, including code generation and mathematical problem-solving OpenAI (2025b); Guo et al. (2025). Pivotal to this success is the implementation of reinforcement learning (RL) to optimize the capacities of LLM to generate step-by-step reasoning Wei et al. (2022) and verify final answers Shao et al. (2024). While multimodal large language models (MLLMs) have achieved impressive performance on established visual reasoning tasks like visual question answering Zhang et al. (2024a); Huang et al. (2025), their core reasoning capabilities remain limited, especially when tasks require processing information-dense images like charts Xu et al. (2024); Chen et al. (2025a), and generating structured outputs like code Xia et al. (2024); Zhao et al. (2025b).

Among these challenges, chart-to-code generation Yang et al. (2024a); Zhang et al. (2024c) stands out as a task of both high complexity and significant practical value. Charts are widely used to communicate scientific findings, business insights, and statistical data, yet their information is often

---

[†] Project leader. [*] Corresponding author.

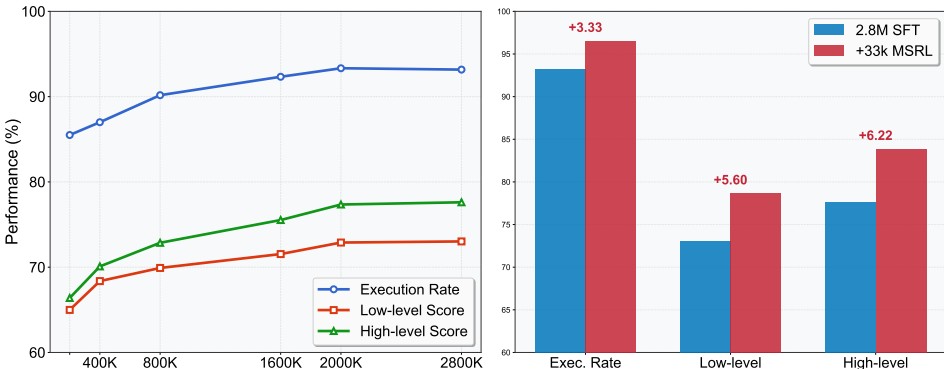

Figure 1: The SFT plateau and RL performance gain from our experiments. The left figure illustrates that scaling SFT data from 200k to 2.8M leads to a performance plateau after exceeding 2M data points. The right figure denotes the performance gain from our proposed MSRL training strategy.

locked in visual formats that are difficult for automated systems to interpret or repurpose. Enabling MLLMs to accurately convert charts into executable code not only advances their visual understanding and structured reasoning capabilities, but also opens up impactful applications, such as assisting researchers in automating the reproduction of scientific visualizations. Although previous works have attempted to solve this task by using methods like Supervised Fine-Tuning (SFT) Zhang et al. (2024b); Zhao et al. (2025b) or Direct Preference Optimization (DPO) Rafailov et al. (2023); Zhang et al. (2025) on synthetic data, their performance remains limited due to the simplistic patterns of the corpus and ineffective generalization strategies. Therefore, a promising direction is to leverage more realistic data and robust strategies to enhance model performance.

In this work, we first systematically investigate the SFT performance plateau through experiments at various data scales. To enable this analysis, we construct the largest training corpus to date for chart-to-code generation, covering 24 distinct chart types and comprising 3 million chart-code pairs. Our two-stage data pipeline curates high-quality tables from arXiv papers and then utilizes LLMs to generate diverse and complex plot codes, ensuring realistic distributions and avoiding monotonous trends. Unlike prior works Ni et al. (2025); Chen et al. (2025b) that apply RL directly after SFT, we establish saturated SFT baselines to isolate the true effectiveness of our proposed multimodal structured reinforcement learning (MSRL) strategy. Through experiments with varying SFT data scales, we demonstrate that simply increasing the volume of SFT data leads to a performance plateau that cannot be overcome by data quantity alone. To break through this plateau, we propose the MSRL training strategy, which incorporates multi-granularity reward functions from both textual and visual aspects. For textual rewards, we standardize code formats in the training corpus and design a comprehensive reward function that verifies code correctness from five key perspectives. Recognizing that text-based rewards alone may overlook overall chart structure, we introduce a visual reward mechanism: generated code is rendered into images and an MLLM evaluates their similarity to the original input chart. By jointly leveraging these multi-granularity rewards, MSRL optimizes both global chart context and fine-grained code details. We evaluate models on various chart-to-code benchmarks Yang et al. (2024a); He et al. (2024). The results demonstrate that it establishes a new state-of-the-art, as it outperforms all previous open-source models and rivals advanced proprietary MLLMs. In summary, the main contributions of this work are as follows:

- We propose an MSRL training strategy that employs a multi-level reward function, which combines a rule-based component for assessing textual code correctness with a model-based evaluator for evaluating the visual fidelity of the rendered chart. Benefits from the MSRL training strategy, our proposed model achieves state-of-the-art (SOTA) performance compared to all other open-source MLLMs.

- We construct the first large-scale chart-to-code training corpus, generated using real-world tables as a data source. After applying several filtering methods, we curate a final dataset of 2.8 million samples for SFT and 33 thousand for RL.

- Through SFT experiments at various data scales, we identify the performance bottleneck, confirming that merely increasing SFT data quantity is insufficient to improve performance.

## 2 RELATED WORK

### 2.1 CHART MLLMs

Chart understanding is a crucial research area encompassing both low-level tasks, such as data extraction Liu et al. (2023), and high-level tasks like question answering Masry et al. (2022) and summarization Kantharaj et al. (2022). Recent work has focused on training MLLMs on extensive, chart-specific datasets to enhance their understanding capabilities Xia et al. (2024); Zhang et al. (2024b); Xia et al. (2023), leading to superior performance across various chart-related tasks. For example, ChartMoE Xu et al. (2024) proposes a mixture-of-experts (MoE) structure for multi-task pretraining. More recently, leveraging reinforcement learning Chen et al. (2025a); Masry et al. (2025); Rodriguez et al. (2025b) to enhance reasoning capacities has garnered significant interest. Both Chart-R1 and BigCharts-R1 optimize MLLMs using Chain-of-Thought (CoT) Wei et al. (2022) reasoning data and reinforcement learning with verified rewards (RLVR) Shao et al. (2024).

### 2.2 REASONING MLLMs

The success of large reasoning models (LRMs) like DeepSeek-R1 Guo et al. (2025) spurs significant research into enhancing LLM reasoning through reinforcement learning with rule-based rewards Shao et al. (2024). This paradigm extends to the vision-language domain, where numerous works aim to improve the long-chain reasoning capabilities of MLLMs Shen et al. (2025); Wang et al. (2025b). Following this trend, initial efforts such as Vision-R1 Huang et al. (2025) and R1-OneVision Yang et al. (2025b) adapt the Group Relative Policy Optimization (GRPO) algorithm with multimodal reasoning data to enable long-form reasoning in VLMs. Subsequent works, including MMEureka Meng et al. (2025) and R1-Zero Liu et al. (2025), further advance this area by introducing improved reinforcement learning strategies for visual long-term reasoning. Recently, approaches like Point-RFT Ni et al. (2025) utilize task-specific thinking frameworks to enhance reasoning capabilities further.

### 2.3 MLLMs FOR CODE

In the rapidly advancing research area of multimodal code generation, foundational work has focused on establishing benchmarks to assess the capabilities of MLLMs. These benchmarks evaluate performance on tasks such as generating execution code for solving visual problems Li et al. (2024); Wang et al. (2025a); Rodriguez et al. (2025a; 2024); Yang et al. (2024b) and HTML code from web-page screenshots Si et al. (2024); Yun et al. (2024); Xiao et al. (2024); Awal et al. (2025). Among these emerging tasks, chart-to-code generation has attracted significant interest due to the complexity of its visual inputs. This task challenges an MLLM to generate plotting code that accurately reproduces the chart image. Recently, several benchmarks have been proposed to evaluate MLLMs in this context, assessing a range of capabilities. For instance, benchmarks such as ChartMimic Yang et al. (2024a), Plot2Code Wu et al. (2025), and ChartX Xia et al. (2024) assess chart-to-code generation capabilities, while others like ChartEdit Zhao et al. (2025a) and ChartM[3] Yang et al. (2025a) evaluate the editing functionalities.

## 3 METHODS

### 3.1 TRAINING CORPUS CONSTRUCTION

While several chart-to-code generation datasets have been proposed, they are generally constrained by a reliance on purely synthetic data and limited scale, resulting in charts that exhibit simplistic trends and lack diversity. To overcome these challenges, we construct our dataset by crawling real-world tables from arXiv papers and then leveraging LLMs to generate the corresponding plotting code. To ensure separation from the ChartMimic benchmark, we exclusively use papers published in 2023 and earlier as our corpus. We ensure data diversity and quality through several strategies, such as using high-quality seed code and filtering out non-executable code.

Through this process, we construct the largest training corpus to date, comprising 3 million chart-code pairs, to investigate the performance limits of SFT. However, previous research Chen et al.

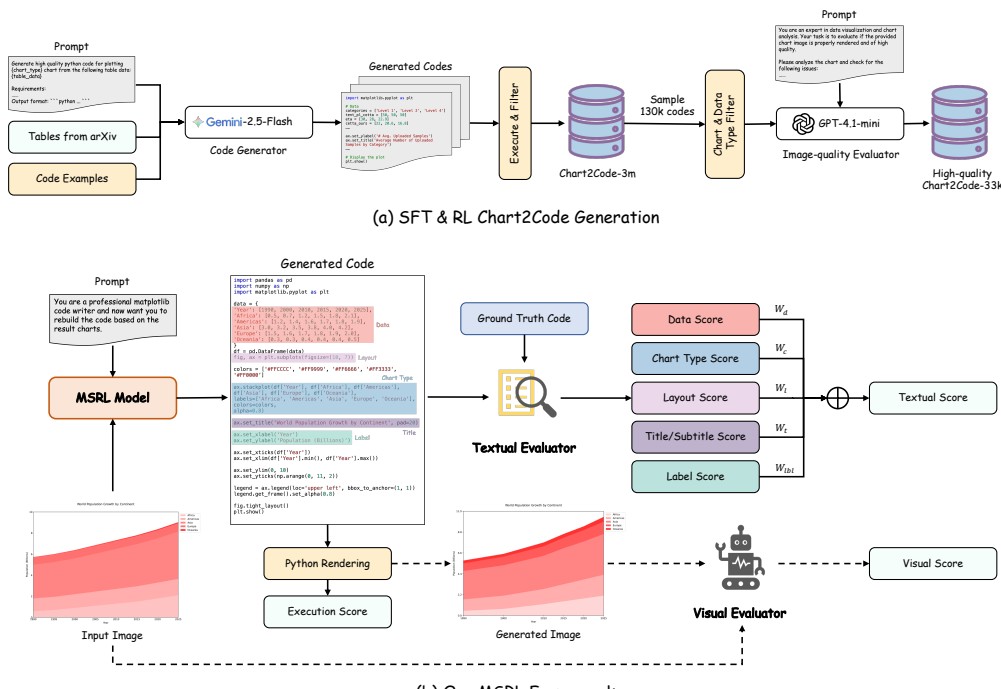

Figure 2: The data generation pipeline and our proposed MSRL framework. (a) Our pipeline prompts Gemini-2.5-Flash with tables from arXiv papers and example codes to generate plotting code. After execution and filtering, the chart-to-code dataset of 3M pairs is obtained. We then sample 130k of these pairs and apply three-stage filters (chart, data, vision) to curate a final, high-quality dataset of 33k examples for RL. (b) The framework of our proposed MSRL strategy. The textual reward is derived from a rule-based evaluation of the generated code across five distinct dimensions. An MLLM quantifies the visual reward based on the rendered image.

(2025a); Huang et al. (2025) shows that separating the SFT and RL training corpora prevents over-fitting to the SFT data format and enhances the exploration capacity of RL. To curate a high-quality dataset for RL, we begin with a candidate corpus of 130 thousand samples from the overall training corpus. These candidate data samples are then subjected to a two-stage filtering strategy. The first filtering stage focuses on the code content, filtering based on chart and data type to ensure code diversity and that the output code is suitable for data extraction. Specifically, the chart types are filtered according to plot functions. To handle complex charts containing multiple chart types, we employ a tree-structured parsing method to identify each type present progressively. The data values are filtered according to the data definition format, retaining only data structured as one-dimensional arrays and non-nested dictionaries to facilitate extraction during the RL stage. This step narrows the candidate corpus to 45 thousand samples. The second stage addresses visual quality, employing GPT-4.1-mini OpenAI (2025a) as a visual LLM judge to filter for high-fidelity chart images. To mitigate potential bias from the MLLM judger, we manually inspect 100 samples that are assigned a score of 0 by the model, and observe over 90% consistency with human judgment. This process yields the final RL training corpus of 33 thousand high-quality samples. The remaining samples are utilized for SFT. The comparisons of our dataset with relevant chart-to-code datasets are listed in Table 1. Our dataset features a completely realistic data source, more challenging multi-chart content, and richer visual diversity enabled by more API types.

## 3.2 SUPERVISED FINE-TUNING

Firstly, we perform SFT on Qwen2.5-VL as the base model using our generated chart-to-code dataset. To explore the effectiveness of data scaling and identify the SFT plateau, we fine-tune our model on six distinct training corpus sizes, created by partitioning our main dataset into sub-sets of 200k, 400k, 800k, 1.6M, 2M and 2.8M samples. The model is trained by minimizing the standard autoregressive objective, which is the negative log-likelihood of the target sequence. As

Table 1: Comparison of existing chart-to-code generation datasets. ✔ indicates that ChartMoE uses both data from real CSV tables and data generated by LLMs. "API types" refers to the number of different Matplotlib API types covered by the dataset.

| Dataset | Realistic data source | Multi-chart | Chart types | Data samples | API types |
|---|---|---|---|---|---|
| ChartLlama | ✗ | ✗ | 10 | 11k | 83 |
| ChartMoE | ✔ | ✗ | <20 | 800k | - |
| Chart2Code | ✗ | ✔ | 15 | 3k | 168 |
| ChartCoder | ✗ | ✗ | 27 | 115k | 187 |
| Ours | ✔ | ✔ | 24 | 3M | 1,555 |

demonstrated in Section 4.2 and Figure 1, merely scaling the SFT data volume ultimately results in a performance plateau. Increasing the SFT dataset from 2M to 2.8M samples results in no further performance gains.

## 3.3 MULTIMODAL STRUCTURED REINFORCEMENT LEARNING

Although SFT can significantly enhance the chart-to-code capabilities of MLLMs, our experiments reveal that this approach ultimately reaches a performance plateau, likely due to the limitation that SFT restricts the model's capacity to restore detailed information. This limitation stems from the core drawback of the SFT objective that it treats every token in the target sequence with uniform importance. Therefore, SFT is ineffective in optimizing plotting code generation, as the plotting code comprises largely of boilerplate snippets such as `plt.plot`, with critical information like specific data values or styling parameters appearing infrequently. Consequently, employing RL with a reward function designed to prioritize the accuracy of such critical content presents a promising approach for enhancing code generation capacity. To this end, we propose a two-stage RL training strategy utilizing multi-granularity reward functions that combine both textual and visual feedback.

**Textual Reward** While the RLVR method provides a powerful mechanism to guide models by measuring the correctness of detailed information, its practical application is challenged by the stylistic diversity of generated code, particularly in data definitions and function calls. This stylistic heterogeneity leads to difficulty in comprehensively extracting and parsing the critical information for reward computation. To mitigate this, we introduce a code normalization step that maps each generated output to a canonical representation before calculating the reward. This process ensures the textual reward function is invariant to syntactic variations, providing a more suitable method for RLVR. Specifically, the textual reward is a granular rule-based accuracy score, which is a weighted average derived from evaluating key aspects of the generated code. We assess data values using soft value matching, chart types with hard string matching, layout with hard value matching and elements like titles and labels via edit distance. The soft value matching allows a relative error tolerance of $\pm5\%$. Separately, we compute an execution reward, which is a binary score indicating whether the generated code can be executed successfully. These two rewards ensure that both the fidelity of critical information and the executability of the code are considered during RL optimization.

**Visual Reward** However, our analysis reveals that purely textual rewards generally focus on fine-grained details, while the overall structure of the rendered chart image is not considered. The chart-to-code task requires generating code that replicates the overall style of charts. To this end, we propose visual reward feedback Gu et al. (2025). Specifically, the generated code is executed to render a chart image, and we then utilize MLLMs to score the visual similarity between the generated chart and the input. This score is then normalized to serve as the visual reward. For the visual reward, we employ Qwen2.5-VL-72B Bai et al. (2025) as the evaluation model, which compares the chart image rendered by the generated code against the ground-truth chart across six key aspects: chart type, layout, text content, data, style, and clarity. The resulting scores are then normalized to form the final reward. Crucially, code that fails to render an image receives a reward of 0. The framework of our proposed MSRL is denoted in Figure 2.

**Two-stage RL** Obtaining visual rewards through model evaluation requires substantial computational time. To ensure training efficiency, our two-stage RL strategy first optimizes the model using only a textual reward. In the second stage, we introduce a hybrid reward, which combines the textual reward with a visual one, to fine-tune the model for visual fidelity using a reduced number of samples. This approach allows us to balance computational cost and model performance, achiev-

Table 2: Evaluation results of various models on the ChartMimic Direct Mimic and ReachQA benchmarks. ‡ denotes the updated result with code_pass constraints reported in its repository.

| Model | Params | ChartMimic | | | ReachQA | | |
|---|---|---|---|---|---|---|---|
| | | Exec.Rate | Low-Level | High-Level | Exec.Rate | Low-Level | High-Level |
| *Proprietary* | | | | | | | |
| GeminiProVision | - | 68.2 | 53.8 | 53.3 | 74.0 | 67.0 | 67.8 |
| Claude-3-opus | - | 83.3 | 60.5 | 60.1 | 89.0 | 51.7 | 61.1 |
| GPT-4V | - | 91.2 | 76.4 | 78.9 | 88.0 | 69.5 | 78.6 |
| GPT-4o | - | 93.2 | **79.0** | 83.5 | 92.8 | 81.8 | 84.0 |
| *Open-Source General-Domain* | | | | | | | |
| Qwen2-VL-7B | 7B | 67.0 | 32.9 | 35.0 | 55.4 | 22.6 | 29.3 |
| Qwen2.5-VL-7B | 7B | 73.2 | 44.6 | 41.6 | 62.2 | 36.9 | 37.6 |
| InternVL2-8B | 8B | 61.8 | 34.4 | 38.9 | 50.8 | 24.1 | 24.2 |
| InternVL2-26B | 26B | 69.3 | 41.4 | 47.4 | 55.4 | 29.0 | 28.8 |
| Qwen2-VL-72B | 72B | 73.3 | 54.4 | 50.9 | 77.2 | 50.0 | 48.1 |
| *Open-Source Chart-Domain* | | | | | | | |
| ChartLlama | 13B | 57.5 | 24.8 | 28.1 | 54.8 | 11.1 | 8.1 |
| TinyChart | 3B | 42.5 | 26.3 | 25.9 | 34.4 | 11.6 | 11.2 |
| ChartVLM-L | 14B | 19.5 | 15.8 | 13.9 | 8.2 | 2.1 | 3.9 |
| Chart2Code | 7B | 62.1 | 42.9 | 33.3 | 63.6 | 52.3 | 49.7 |
| ChartCoder | 7B | 91.4 | 72.5‡ | 74.0 | 83.8 | 67.9 | 69.4 |
| MSRL-SFT | 7B | 93.2 | 73.0 | 77.6 | 92.2 | 78.6 | 80.0 |
| MSRL | 7B | **96.5** | 78.6 | **83.8** | **98.2** | **86.1** | **89.9** |

ing strong visual quality without excessive resource consumption. To properly balance the different training objectives, the total reward $R$ for a completed trajectory is calculated as a weighted sum of the following three components:

$$R = w_t R_{\text{text}} + w_v R_{\text{vis}} + w_e R_{\text{exec}} \quad (1)$$

where $w_t$, $w_v$, and $w_e$ are hyperparameters that balance the contribution of each component. We adopt the Group Relative Policy Optimization (GRPO) Shao et al. (2024) algorithm for RL. GRPO optimizes policy by leveraging group-wise relative advantages among sampled responses, which is well-suited for our customized reward designs of charts.

# 4 EXPERIMENT

## 4.1 BASELINES AND BENCHMARKS

We evaluate MSRL against three categories of state-of-the-art models. The first category comprises general-domain, open-source Multimodal Large Language Models (MLLMs): InternVL2 (8B, 26B) Chen et al. (2024b), Qwen2-VL (7B, 72B) Wang et al. (2024), and Qwen2.5-VL-7B Bai et al. (2025). The second includes proprietary models: GeminiProVision Team et al. (2023), Claude-3-opus Anthropic (2024), GPT-4V OpenAI (2023), and GPT-4o OpenAI (2024). The third category consists of chart-specific MLLMs: ChartLlama Han et al. (2023), Tinychart Zhang et al. (2024b), ChartVLM Xia et al. (2024), Chart2Code Zhang et al. (2025) and ChartCoder Zhao et al. (2025b).

All models were evaluated in a zero-shot setting on two distinct benchmarks: ChartMimic Yang et al. (2024a) and ReachQA He et al. (2024). For ReachQA, we adopt the evaluation setting of Chart2Code Zhang et al. (2025), using the 500 plotting scripts from its test set. For ChartMimic, we utilize the 600 examples from the latest version of the Direct Mimic task.

## 4.2 MAIN RESULTS

Our proposed MSRL establishes a new state-of-the-art among all open-source MLLMs, as detailed in Table 2. On the ChartMimic benchmark Yang et al. (2024a), it achieves an execution rate of

Table 3: Detailed results of low-level scores on the ChartMimic Direct Mimic and ReachQA benchmarks. The ChartCoder performance metrics have been corrected due to a flawed evaluation setup.

| Model | Params | ChartMimic | | | | ReachQA | | | |
|---|---|---|---|---|---|---|---|---|---|
| | | Text | Layout | Type | Color | Text | Layout | Type | Color |
| GPT-4o | - | **81.5** | 89.8 | **77.3** | 67.2 | 84.4 | 91.2 | 81.3 | 70.5 |
| Qwen2-VL-72B | 72B | 43.2 | 80.5 | 54.6 | 39.4 | 41.0 | 52.2 | 59.1 | 47.8 |
| Qwen2.5-VL-7B | 7B | 37.7 | 65.6 | 42.6 | 32.4 | 29.6 | 40.9 | 44.2 | 32.8 |
| ChartCoder[‡] | 7B | 65.9 | 85.4 | 72.3 | 66.4 | 58.2 | 80.0 | 70.4 | 63.2 |
| MSRL | 7B | 78.0 | **93.0** | 75.3 | **68.2** | **87.5** | **96.6** | **83.1** | **77.2** |

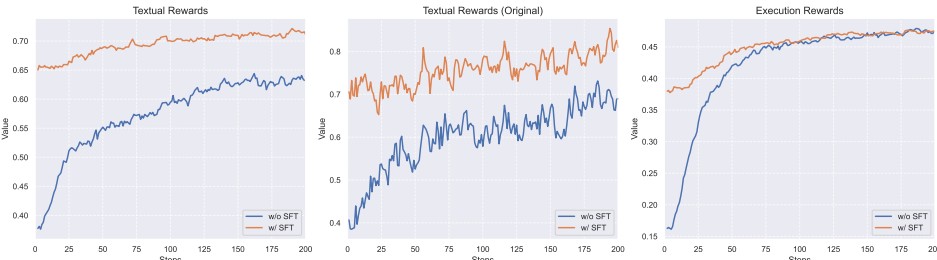

Figure 3: Comparison of textual reward and execution rate changes between baseline and SFT models during the RL stage.

96.5, a low-level score of 78.6, and a high-level score of 83.8. Notably, its performance is not only superior to its open-source models but also comparable to that of GPT-4o, a significantly larger proprietary model. On the ChartMimic benchmark, MSRL outperforms ChartCoder, the previous leading Chart-domain MLLM specifically designed for chart-to-code generation. Unlike other works that yield suboptimal performances from fine-tuning open-source models, our initial MSRL-SFT model already establishes a new state-of-the-art among open-source models. However, MSRL-SFT still falls short of GPT-4o on both low-level and high-level performance metrics. Building upon this strong foundation, the MSRL strategy further boosts the performance, especially in high-level scores, demonstrating the effectiveness of our integrated approach. The performance gap between SFT and RL models is emphasized in Figure 1.

We also conduct a detailed evaluation of low-level performance metrics, with the results presented in Table 3. Notably, MSRL outperforms all open-source models across low-level evaluation metrics and surpasses GPT-4o on the majority of evaluation metrics, which shows the strong capacity of MSRL in chart-to-code generation. As the GPT-4o is much larger than MSRL, the results demonstrate the effectiveness of our proposed methods.

**SFT Plateau** To investigate and demonstrate the effectiveness and efficiency of RL, we utilize various data scales for SFT to explore the upper limit of it. As illustrated in Figure 1, the results reveal a clear trend, whereby model performance improves sharply as the data size increases to 400k samples, followed by a period of slower growth up to the 1.6M samples. Beyond this data scale, performance saturates and reaches a distinct plateau, showing negligible growth with any further increase in data scale. Based on this trend, even if the data is multiplied, the benefits brought by SFT will be minimal. Applying our proposed MSRL method yields significant performance gains across all metrics. On the ChartMimic benchmark, we observe an average improvement of 6%, with notable increases in execution rate as well as in both low-level and high-level scores.

**RL Plateau** To investigate whether RL training exhibits a performance plateau, we present the performance curves of the two-stage RL training with the increase of data size, as shown in Figure 4. In the first stage using only textual rewards, the model breaks through the SFT plateau and achieves a significant performance improvement of approximately 5%, converging at a maximum of 22k training samples. In the second stage with multimodal rewards, the model's performance is further improved by about 1.5%, and almost converges at a maximum of 11k training samples. Our experiments effectively capture the performance plateau and improvements within a practical data size range, providing strong evidence for the effectiveness of our two-stage RL strategy.

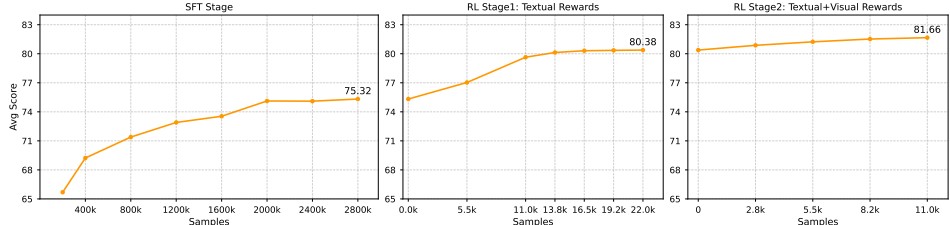

Figure 4: Performance curve of MSRL with increasing data scale. From left to right, the curves correspond to the SFT stage, RL stage 1, and RL stage 2. Avg Score denotes the average of the Low- and High-level scores in ChartMimic.

## 4.3 ABLATION STUDY

**SFT/RL Ablation** To understand the impact of different training settings, we conduct ablation studies on both the SFT and RL training stages. In this ablation study, we use only textual rewards to ensure the results are not influenced by our improved RL algorithm, allowing us to demonstrate the interplay between SFT and RL. Using Qwen2.5-VL-7B-Instruct as the baseline model, the results are presented in Table 4. As the RL-only model

Table 4: Ablation studies on SFT and RL stages. For two RL experiments, only a textual reward is used.

| SFT | RL | ChartMimic | | |
|---|---|---|---|---|
| | | Exec.Rate | Low-Level | High-Level |
| | | 73.16 | 44.58 | 41.55 |
| ✓ | | 93.17 | 73.02 | 77.61 |
| | ✓ | 93.83 | 65.60 | 62.29 |
| ✓ | ✓ | **97.00** | **78.06** | **82.69** |

underperforms on low-level and high-level scores compared to the SFT-only approach, only the execution rate is slightly higher due to the inclusion of an execution-based reward. Furthermore, the significantly larger performance gain from RL on a suboptimal baseline, compared to an SFT model, indicates that this improvement is partly attributable to the unsaturated state of baseline models. The performance gain on a saturated SFT model demonstrates that our proposed multi-granularity reward function can surpass the upper limit of SFT, achieving significantly superior performance. Figure 3 compares the reward and execution rate for the baseline and SFT models during the RL training, showing that applying RL to the SFT model achieves a much higher textual reward.

Table 5: Comparison of single-stage and two-stage RL training with different reward strategies. "T" and "V" represent textual and visual reward, respectively. All training is conducted on H800 GPUs.

| Rewards | Samples | ChartMimic | | | GPU Hours |
|---|---|---|---|---|---|
| | | Exec.Rate | Low-Level | High-Level | |
| - | - | 93.17 | 73.02 | 77.61 | - |
| (T, -) | (22k, -) | 97.00 | 78.06 | 82.69 | 240 |
| (V, -) | (22k, -) | 97.37 | 79.13 | 84.01 | 1344 |
| (T, T + V) | (22k, 11k) | 97.50 | 79.50 | 83.81 | 912 |
| (T, T + V) | (22k, 5.5k) | 96.50 | 78.62 | 83.83 | 576 |

**Reward and RL Training Strategy Ablation** To thoroughly evaluate the effects of reward design and RL training strategies, we conduct ablation experiments comparing textual rewards, visual rewards, and their combinations. As shown in Table 5, both reward types improve model performance, but differ notably in resource consumption and efficiency. Visual rewards provide notable enhancements in code details and visual fidelity, but require more computational resources due to the overhead of image rendering and model evaluation. In contrast, textual rewards achieve competitive performance with much lower resource requirements. To balance efficiency and effectiveness, we adopt a two-stage RL strategy. The ablation results yield several important insights: 1) Both textual and visual rewards lead to significant improvements over the SFT model, with visual rewards being more effective for visual fidelity. 2) The high resource consumption of visual rewards highlights the importance of balancing performance and efficiency. 3) The two-stage RL strategy achieves the trade-off by first using low-cost textual rewards, followed by high-cost visual rewards with a limited number of samples. Based on these findings, we empirically report the two-stage training strategy with 50% of the training samples in our experiments, as it offers a better trade-off between performance and consumption efficiency.

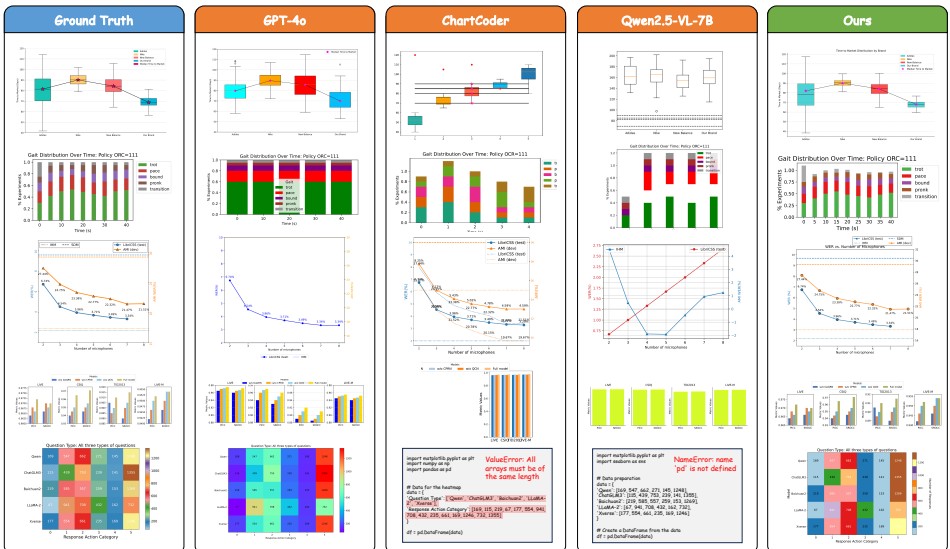

Figure 5: Showcasing charts generated by MSRL compared to proprietary and open-source MLLMs. The first column shows the ground-truth chart images. Columns 2–4 display charts rendered from the code generated by GPT-4o, ChartCoder, and Qwen2.5-VL-7B, while the last column shows charts rendered by our MSRL model.

**Generalization to Unseen Plotting Libraries** To evaluate whether MSRL can transfer its capabilities to other plotting libraries, we collect 107 Seaborn and 150 Plotly images from their official websites. Since we only use Matplotlib-style code to generate charts, the new test data are out-of-domain. We perform inference with both Qwen2.5-VL-7B and MSRL on these two small test sets, calculating execution rates and high-level scores according to the ChartMimic protocol. As shown in Table 6, the results demonstrate that MSRL exhibits generalization to unseen Seaborn-and Plotly-style images.

Table 6: Performance of Qwen2.5-VL-7B and MSRL on Seaborn and Plotly test sets.

| Model | Seaborn | | Plotly | |
|---|---|---|---|---|
| | Exec.Rate | High-Level | Exec.Rate | High-Level |
| Qwen2.5-VL-7B | 69.2 | 25.7 | 62.7 | 22.6 |
| MSRL | **85.1** | **30.5** | **90.0** | **35.9** |

## 4.4 VISUALIZATION

To provide a qualitative perspective on performance, we showcase several illustrative examples. As visually demonstrated in Figure 5, the charts generated by MSRL exhibit a markedly higher fidelity to the ground truth when compared with leading open-source MLLMs, even GPT-4o. In particular, the data values and layouts of charts generated by MSRL are significantly more precise.

## 5 CONCLUSION

In this work, we investigate the performance plateau of Supervised Fine-Tuning (SFT) in chart-to-code generation and propose a Multimodal Structured Reinforcement Learning (MSRL) strategy to overcome it. We begin by constructing a 3M-pair training corpus from real-world tables to systematically demonstrate that merely increasing SFT data leads to diminishing returns. To address this, we introduce the MSRL, which optimizes the saturated SFT model using a two-stage training process guided by a multi-granularity reward function that assesses both textual and visual correctness. Our approach achieves new state-of-the-art performance on standard benchmarks, surpassing all open-source models and rivaling powerful proprietary ones. We make significant contributions to the field, including the curation of a large-scale dataset for a definitive empirical analysis of the SFT performance bottleneck, and the development of the MSRL framework.

## 6 REPRODUCIBILITY STATEMENT

For datasets, we provide a detailed description of the data generation process in Section 3.1 of the paper, along with dataset statistics and examples in Appendix A, and the exact prompts used for dataset construction in Appendix E. For code implementation, we include the SFT and RL training code for MSRL in the supplementary materials, which contains complete training scripts and implementations of the multimodal reward functions.

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

## A CHART2CODE DETAILS

We count the quantity and distribution of 24 chart types used in the SFT and RL stages respectively, as detailed in Table A. The distribution among the various types is relatively balanced. For the RL data, we filtered out two chart types, Graph and Funnel, which are not suitable for textual reward calculation. Furthermore, we present three visualization examples from the Chart2Code dataset, with increasing data density and complexity, as shown in Figures G, H and I.

Table A: The detailed chart types and corresponding quantities used in the SFT and RL phases.

| Split | Bar | Line | ErrorBar | Heatmap | Box | Scatter | Histogram | Radar | 3D |
|---|---|---|---|---|---|---|---|---|---|
| SFT | 131,623 | 131,596 | 130,818 | 123,795 | 125,541 | 127,612 | 132,180 | 124,160 | 128,744 |
| RL | 2,669 | 2,743 | 1,067 | 1,984 | 756 | 2,639 | 1,510 | 776 | 1,699 |

| Split | Pie | ErrorPoint | Violin | Area | Bubble | Multi-axes | Ring | Rose | Treemap |
|---|---|---|---|---|---|---|---|---|---|
| SFT | 125,812 | 123,774 | 120,180 | 120,038 | 127,207 | 116,438 | 105,912 | 118,047 | 119,176 |
| RL | 1,446 | 2,277 | 900 | 1,558 | 1,093 | 1,691 | 960 | 674 | 518 |

| Split | Bar_num | Contour | Density | Quiver | Graph | Funnel | Total |
|---|---|---|---|---|---|---|---|
| SFT | 128,333 | 123,461 | 130,769 | 128,987 | 108,150 | 1,152 | 2,853,505 |
| RL | 2,757 | 1,040 | 1,899 | 1,700 | - | - | 34,356 |

## B  IMPLEMENTATION DETAILS

We utilize 2.8M chart2code data for SFT and 33k curated high-quality chart2code data for RL. MSRL-SFT employs Qwen2.5VL-7B-Instruct as the baseline model and LLaMA-Factory Zheng et al. (2024) for SFT. The SFT stage is trained for one epoch with a learning rate of 1e-5 and a batch size of 32. The training process takes 60 hours on 16 H800 GPUs. Built upon the SFT model, MSRL undergoes a two-stage RL process using the MM-EUREKA framework Meng et al. (2025). MSRL consists of two RL stages. Stage one utilizes 22k data pairs and a textual-only reward ($w_t = 1$, $w_v = 0$, $w_e = 0.5$). Stage two uses the remaining 11k pairs with a hybrid reward ($w_t = 0.5$, $w_v = 0.5$, $w_e = 0.5$). The textual reward components are weighted as $w_d = 0.4$, $w_c = 0.3$, $w_l = 0.1$, $w_t = 0.1$, $w_{lbl} = 0.1$. We deploy Qwen2.5-VL-72B-Instruct as the evaluation model with max_model_len set to 12,288 for calculating visual rewards. For both RL stages, we train for one episode with a learning rate of 1e-6 and a batch size of 128. We set the generation temperature to 1, generate 8 rollouts per sample, and omit the KL divergence from the loss computation. The first and second stages required 10 and 24 hours of training, respectively, on 24 H800 GPUs.

## C  MORE ABLATION STUDIES

**The Use of Visual Evaluator** To verify that the improvement in performance results from effective visual reward design rather than overfitting to the preferences of the VLLM judger, we conduct an ablation experiment to observe the distribution of sample scores when the visual evaluator changes. Specifically, we randomly sample 200 instances from the RL data and use the MSRL-SFT model to perform chart-to-code prediction, rendering the resulting Python code into images. Following the ChartMimic scoring guidelines, human ratings of rendered samples serve as the gold scores. Next, we use GPT-4.1, Qwen2.5-VL-72B, Qwen2.5-VL-7B, and InternVL3-38B to score the images. Additionally, we compute the visual similarity between the ground-truth and rendered images using DINO-L. We calculate the Pearson correlation coefficients between the scores from these models and the human scores, and plot the distributions of model scores versus human scores. As shown in the Figure A, GPT-4.1 achieves the highest correlation with human judgment and has the most similar score distribution, followed by Qwen2.5-VL-72B, while Qwen2.5-VL-7B performs the worst. Considering both performance and budget, we select Qwen2.5-VL-72B as the visual evaluator, as it fairly reflects model capability and prevents reward hacking.

In addition, we introduce two vision models, InternViT Chen et al. (2024a) and DINOv2-L Oquab et al. (2023), as evaluators for visual rewards, calculating the visual similarity between ground-truth and generated image features. As shown in Table B, these vision models yield improved performance, but there remains a substantial gap compared to models employing VLMs as visual evaluators. This indicates that vision models trained on image-text matching assess similarity at a coarse level and do not adequately capture fine-grained differences in chart elements. Compared to Qwen2.5-VL-72B, smaller vision models such as DINO-L exhibit greater discrepancies in score distributions relative to human judgment.

**GRPO vs DPO** Intuitively, GRPO is naturally more suitable for chart-to-code tasks than DPO. The degree to which code restores a chart can be quantitatively measured by a set of well-defined rules, making it ideal for designing the reward function in GRPO. In contrast, DPO learns from preference

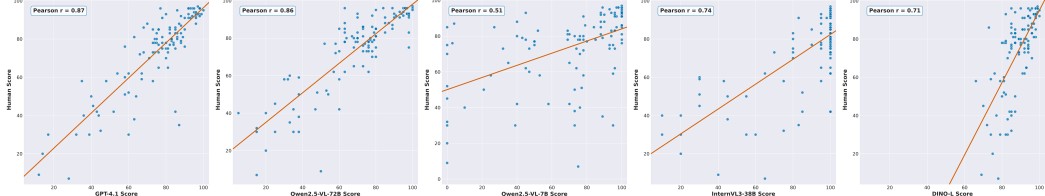

Figure A: Scatter plots of model scores versus human gold scores for five visual evaluators (GPT-4.1, Qwen2.5-VL-72B, Qwen2.5-VL-7B, InternVL3-38B, and DINO-L). Pearson $r$ denotes the Pearson correlation coefficient between model and human scores.

Table B: The ablation study of visual evaluators. "-" denotes the SFT model. All experiments exclude textual rewards.

| Visual Evaluator | ChartMimic | |
| --- | --- | --- |
| | Low-Level | High-Level |
| - | 73.0 | 77.6 |
| InternViT | 75.1 | 78.1 |
| DINOv2-L | 75.5 | 79.8 |
| Qwen2.5-VL-72B | **77.8** | **81.9** |

pairs, and it is challenging to clearly define what constitutes "bad" code in chart-to-code tasks, which can lead to ambiguity in the model's optimization direction. To ensure a thorough evaluation, we conduct an ablation study comparing GRPO and DPO. Since DPO requires the construction of positive and negative sample pairs, we follow the Chart2Code Zhang et al. (2025) and randomly perturb 2-3 attributes of the chart code to generate negative samples using GPT-4.1, resulting in an RL dataset of comparable size to GRPO for DPO training on the SFT model. As shown in Table C, the performance improvement of DPO is significantly less than that of GRPO. This indicates both the superiority of the GRPO algorithm and the effectiveness of our multimodal structured reward design and two-stage curriculum learning.

Table C: The ablation study of RL optimizers. "-" denotes the SFT model.

| RL Optimizer | ChartMimic | |
| --- | --- | --- |
| | Low-Level | High-Level |
| - | 73.0 | 77.6 |
| DPO | 74.0 | 78.9 |
| GRPO | **78.6** | **83.8** |

**Comparison with Existing Datasets** To highlight the value of our synthetic data, we compare the results of using our data and open-source data (i.e., ChartCoder) in both the SFT and RL stages, as shown in Table D. The results show that models trained with our data achieve better performance in both stages. On the one hand, our sufficient data enable SFT to reach its full potential. On the other hand, our charts are generated from real tabular data, featuring more challenging multi-chart content and greater visual diversity, which are key factors contributing to the effectiveness of our dataset.

# D QUALITATIVE ANALYSIS

Figures E and F show two examples of generated code from different models and their corresponding rendered images. Compared to the initial Qwen2.5-VL-7B model, both MSRL-SFT and MSRL show significant improvements in chart type recognition, numerical extraction, and color identification. Compared to MSRL-SFT, MSRL performs better in chart type recognition and clarity. In Figure E, MSRL-SFT incorrectly identifies the line chart as a scatter plot, displays incorrect legends, and has overlapping issues with image boundaries. In Figure F, MSRL-SFT has layout syntax errors, where only the last subplot is successfully displayed out of three subplots. MSRL's results are more consistent with the original images, though it still has some minor issues in detail. For

Table D: Performance comparison with the ChartCoder dataset. We use Qwen2.5-VL-7B and MSRL-SFT as base models for SFT and RL for fine-tuning on the two datasets, respectively.

| Stage | Dataset | ChartMimic | |
| --- | --- | --- | --- |
| | | Low-Level | High-Level |
| SFT | ChartCoder | 68.5 | 68.0 |
| | Ours | **73.0** | **77.6** |
| RL | ChartCoder | 76.5 | 81.2 |
| | Ours | **78.6** | **83.8** |

example, in Figure E, MSRL arbitrarily adds a title and scatter data points to the subcategory "Ours (LoSA)". In Figure F, while MSRL successfully renders three subplots, the layout recognition is not sufficiently accurate, leaving room for further improvement.

## E  PROMPT TEMPLATE

To enhance transparency and reproducibility, we provide the exact prompts used for dataset generation and visual reward feedback.

Figure B shows the prompt used for code generation. We use real table data as input, select one chart type from 24 predefined chart types, and sample code examples of the selected chart type to generate plotting code. Specifically, for chart types Pie, Ring, and Treemap, we require the model to display actual values on the image, as these chart types typically only visualize percentage values in their code, making it impossible to read the true values from the image.

Figure C shows the prompt used for evaluating image quality. When processing RL data, we use this prompt to filter out visually low-quality samples, such as those with overlapping text or elements.

Figure D shows the prompt used to calculate visual rewards. Following ChartMimic Yang et al. (2024a), we assess the match between ground truth and model predictions across 6 dimensions, with this score normalized as the visual score and incorporated into the overall reward computation.

---

**Prompt Template of Code Generation**

Generate high quality python code for plotting {chart_type} chart from the following table data:
{table_data}

Requirements:
The code must present table data in a reasonable way.
The code example of {chart_type} chart (given in JSON format) is:
{code_example}

You must not be limited by the code sample and draw different styles of dials.
The generated code should not be too complicated and all text elements (labels, titles, legends) must be fully visible without overlap or truncation.
Pie/Ring/Treemap chart visualization: always display the actual numerical values on each segment. Percentages are optional, but values must be clearly visible.
IMPORTANT: Generate only ONE figure with all necessary information. If multiple plots are needed, use subplots (plt.subplots) to arrange them in a single figure.
Output format: ```python ... ```

---

Figure B: Prompt template for Chart2Code code generation.

## F  THE USE OF LARGE LANGUAGE MODELS (LLMS)

In this work, we utilized LLMs primarily as writing assistance tools to polish language and improve clarity. They helped refine sentence structures and enhance readability throughout the manuscript. The core research ideas, experimental design, implementation, and analysis were all conducted by the authors, with LLMs serving only as editorial aids.

---

**Prompt Template of Image-quality Evaluation**

You are an expert in data visualization and chart analysis. Your task is to evaluate if the provided chart image is properly rendered and of high quality.

Please analyze the chart and check for the following issues:
1. Rendering problems (missing elements, incorrect display)
2. Data anomalies (outliers that don't make sense, inconsistent scales)
3. Truncated or cut-off text or elements
4. Overlapping text or elements that reduce readability
5. Poor color choices that make the chart hard to read
6. Missing labels or legends that are necessary for understanding
7. Any other quality issues that affect the chart's usefulness

After your analysis, provide a single score:
- Score 1: The chart is properly rendered, data appears normal, and there are no significant quality issues.
- Score 0: The chart has one or more significant quality issues that affect its usefulness.

Respond with ONLY the score (1 or 0) and nothing else.

---

Figure C: Prompt template for image-quality evaluation during RL data filtering.

---

**Prompt Template of Visual Reward Feedback**

You are an excellent judge at evaluating visualization chart plots. The first image (reference image) is created using ground truth matplotlib code, and the second image (AI-generated image) is created using matplotlib code generated by an AI assistant. Your task is to score how well the AI-generated plot matches the ground truth plot.

### Scoring Methodology:
The AI-generated image's score is based on the following criteria, totaling a score out of 100 points:

1. **Chart Types (20 points)**
   Does the AI-generated image include all chart types present in the reference image
   (e.g., line charts, bar charts, etc.)?
2. **Layout (10 points)**
   Does the arrangement of subplots in the AI-generated image match the reference image
   (e.g., number of rows and columns)?
3. **Text Content (20 points)**
   Does the AI-generated image include all text from the reference image
   (e.g., titles, annotations, axis labels), excluding axis tick labels?
4. **Data (20 points)**
   How accurately do the data trends in the AI-generated image resemble those in the original image
   and is the number of data groups the same as in the reference image?
5. **Style (20 points)**
   Does the AI-generated image match the original in terms of colors (line colors, fill colors, etc.),
   marker types (point shapes, line styles, etc.), legends, grids, and other stylistic details?
6. **Clarity (10 points)**
   Is the AI-generated image clear and free of overlapping elements?

### Evaluation:
Compare the two images head to head and provide a detailed assessment. Use the following format for your response:

---
Comments:
- Chart Types: ${your comment and subscore}
- Layout: ${your comment and subscore}
- Text Content: ${your comment and subscore}
- Data: ${your comment and subscore}
- Style: ${your comment and subscore}
- Clarity: ${your comment and subscore}

Score: ${your final score out of 100}
---

Please use the above format to ensure the evaluation is clear and comprehensive.

---

Figure D: Prompt template for visual reward feedback during the RL training.

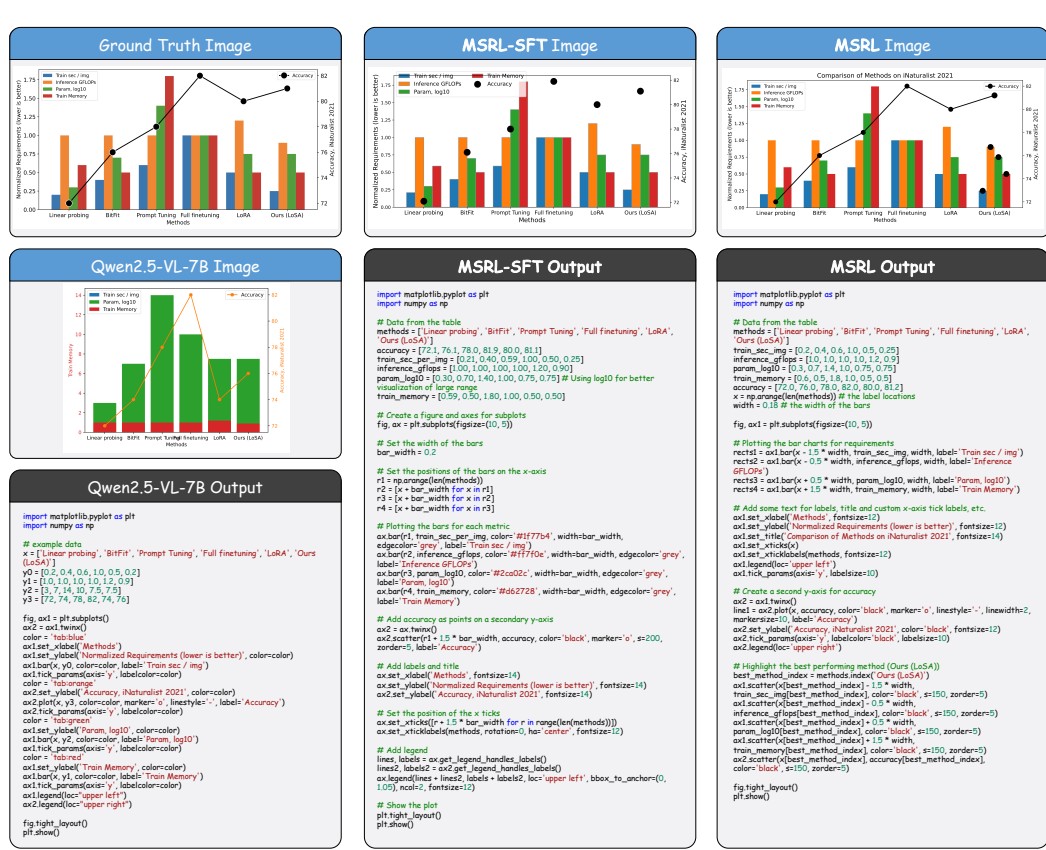

Figure E: An example of comparing the code generated by different models with rendered images.

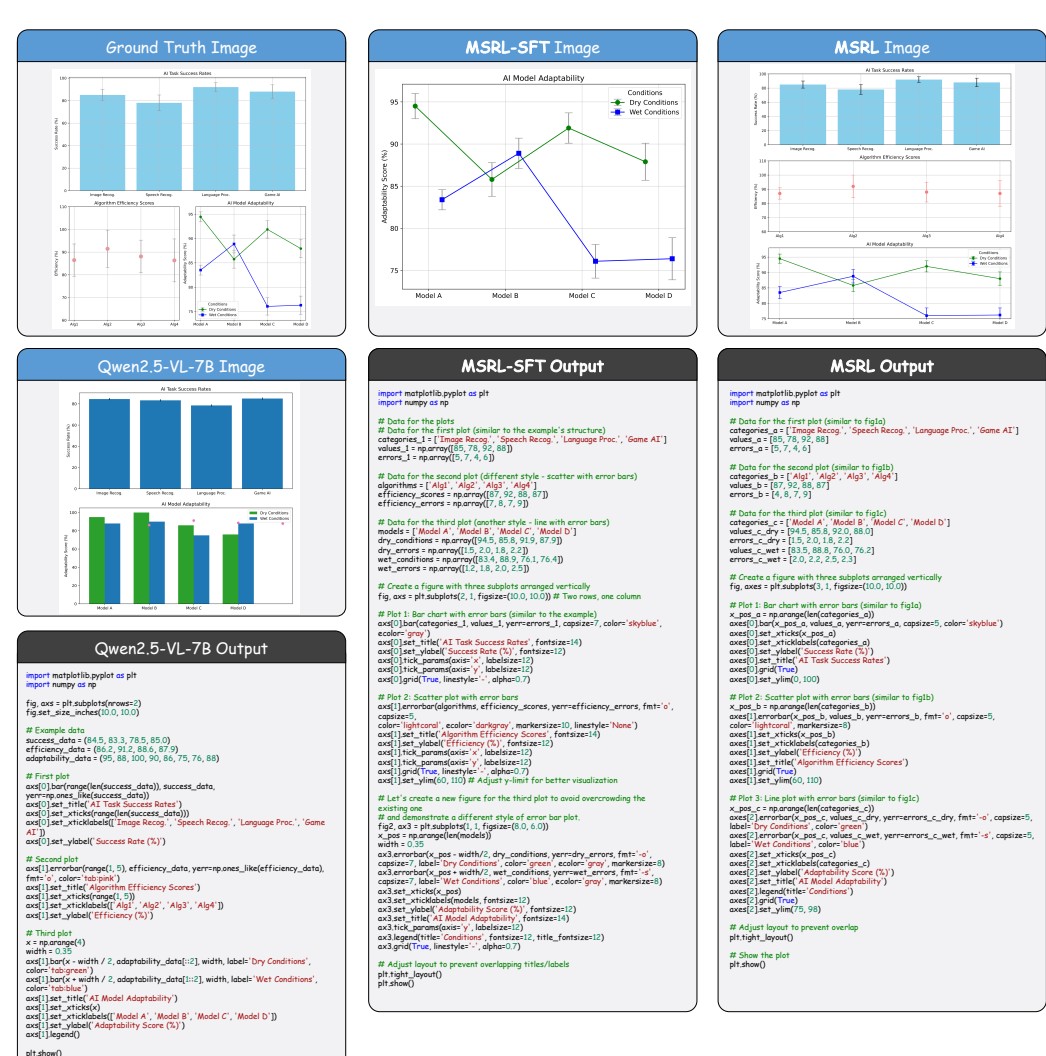

Figure F: An example of comparing the code generated by different models with rendered images.

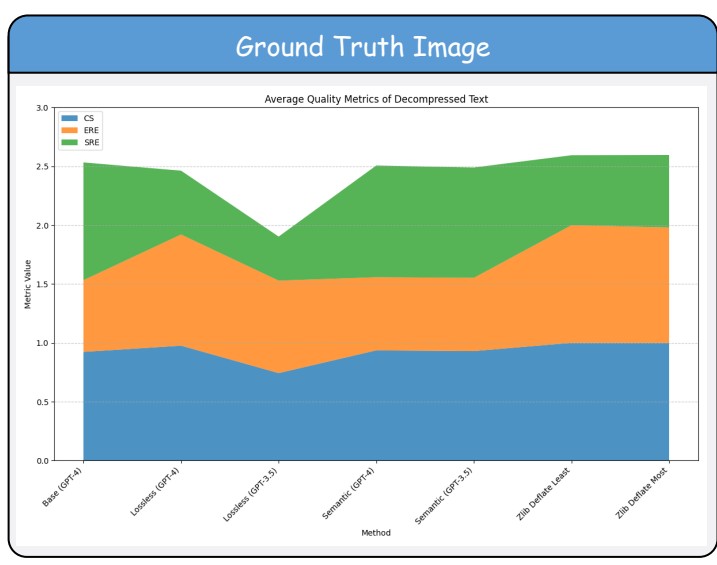

Figure G: A case of chart-code pairs for area charts in our Chart2Code dataset.

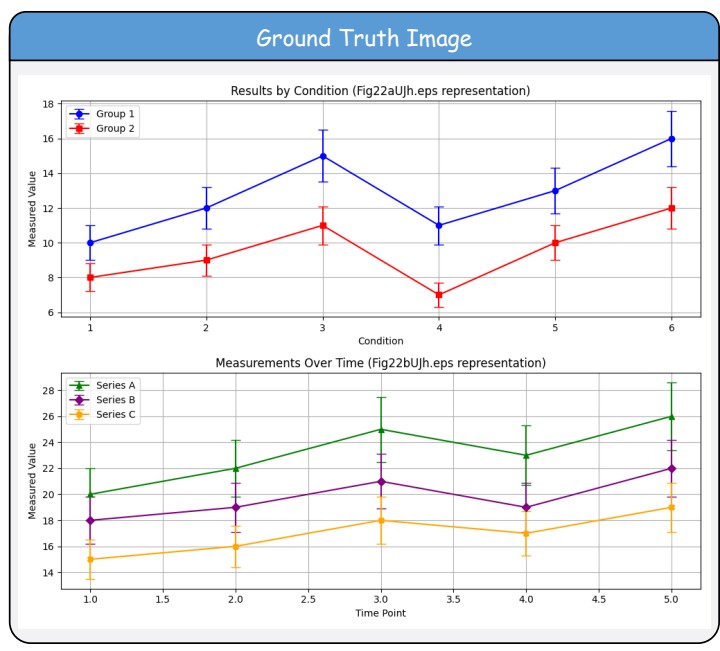

Figure H: A case of chart-code pairs for errorbar charts in our Chart2Code dataset.

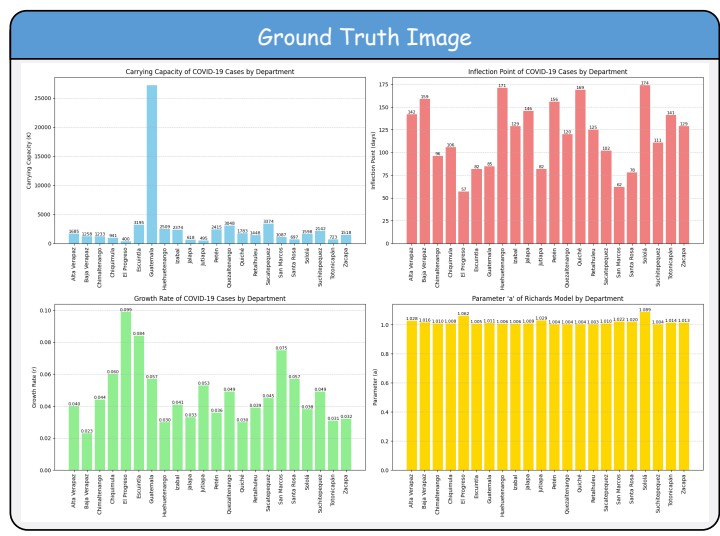

```python
import matplotlib.pyplot as plt
# Data from the table
departments = [
'Alta Verapaz', 'Baja Verapaz', 'Chimaltenango', 'Chiquimula', 'El Progreso',
'Escuintla', 'Guatemala', 'Huehuetenango', 'Izabal', 'Jalapa', 'Jutiapa',
'Petén', 'Quezaltenango', 'Quiché', 'Retalhuleu', 'Sacatepequez', 'San Marcos',
'Santa Rosa', 'Sololá', 'Suchitepequez', 'Totonicapán', 'Zacapa'
]
a = [1.028, 1.016, 1.010, 1.008, 1.062, 1.005, 1.011, 1.006, 1.006, 1.009, 1.029, 1.004, 1.004, 1.004, 1.003, 1.010,
1.022, 1.020, 1.089, 1.004, 1.014, 1.013]
growth_rate = [0.040, 0.023, 0.044, 0.060, 0.099, 0.084, 0.057, 0.030, 0.041, 0.033, 0.053, 0.036, 0.049,
0.030, 0.039, 0.045, 0.075, 0.057, 0.038, 0.049, 0.031, 0.032]
carrying_capacity = [1685, 1258, 1233, 941, 400, 3195, 27230, 2509, 2374, 618, 495, 2415, 3048, 1783,
1448, 3374, 1087, 697, 1598, 2142, 723, 1518]
inflection_point = [142, 159, 96, 106, 57, 82, 85, 171, 129, 146, 82, 156, 120, 169, 125, 102, 62, 78, 174, 111,
141, 129]

# Create subplots
fig, axes = plt.subplots(nrows=2, ncols=2, figsize=(18, 12))
axes = axes.flatten()
# Plotting Carrying Capacity
ax1 = axes[0]
bars_cc = ax1.bar(departments, carrying_capacity, color='skyblue')
ax1.set_ylabel('Carrying Capacity (K)')
ax1.set_title('Carrying Capacity of COVID-19 Cases by Department')
ax1.tick_params(axis='x', rotation=90)
ax1.grid(axis='y', linestyle='--', alpha=0.7)
for bar in bars_cc:
        height = bar.get_height()
        if height < 10000:  # Only show values for smaller bars to avoid clutter
                ax1.text(bar.get_x() + bar.get_width()/2., height,
                '%d' % int(height), ha='center', va='bottom', fontsize=8)
# Plotting Inflection Point
ax2 = axes[1]
bars_ip = ax2.bar(departments, inflection_point, color='lightcoral')
ax2.set_ylabel('Inflection Point (days)')
ax2.set_title('Inflection Point of COVID-19 Cases by Department')
ax2.tick_params(axis='x', rotation=90)
ax2.grid(axis='y', linestyle='--', alpha=0.7)
for bar in bars_ip:
        height = bar.get_height()
        ax2.text(bar.get_x() + bar.get_width()/2., height,
        '%d' % int(height), ha='center', va='bottom', fontsize=8)
# Plotting Growth Rate
ax3 = axes[2]
bars_gr = ax3.bar(departments, growth_rate, color='lightgreen')
ax3.set_ylabel('Growth Rate (r)')
ax3.set_title('Growth Rate of COVID-19 Cases by Department')
ax3.tick_params(axis='x', rotation=90)
ax3.grid(axis='y', linestyle='--', alpha=0.7)
for bar in bars_gr:
        height = bar.get_height()
        ax3.text(bar.get_x() + bar.get_width()/2., height,
        '%.3f' % height, ha='center', va='bottom', fontsize=8)
# Plotting 'a' parameter
ax4 = axes[3]
bars_a = ax4.bar(departments, a, color='gold')
ax4.set_ylabel('Parameter (a)')
ax4.set_title('Parameter \'a\' of Richards Model by Department')
ax4.tick_params(axis='x', rotation=90)
ax4.grid(axis='y', linestyle='--', alpha=0.7)
for bar in bars_a:
        height = bar.get_height()
        ax4.text(bar.get_x() + bar.get_width()/2., height,
        '%.3f' % height, ha='center', va='bottom', fontsize=8)
plt.tight_layout()
plt.show()
```

Figure I: A case of chart-code pairs for bar_num charts in our Chart2Code dataset.

