# OpenReview forum: "Breaking the SFT Plateau: Multimodal Structured Reinforcement Learning for Chart-to-Code Generation"
_ICLR.cc/2026/Conference — ICLR 2026 Poster_

### Official Review · Reviewer_npBT · 2025-10-19

**Soundness:** 3
**Presentation:** 2
**Contribution:** 2
**Rating:** 4
**Confidence:** 4

**Summary:**

The paper focuses on chart-to-code generation. The authors built a large chart-code corpus and found that SFT hits a performance plateau when training data is scaled, with more data bringing little improvement. To address this, the authors used MSRL to fine-tune MLLMs and design task-specific rewards. The MSRL method achieved better performance than SFT. Moreover, the trained MLLMs outperform most baselines.

**Strengths:**

1. The methods of data synthesis and reinforcement learning are intuitive and interesting, providing a certain basis for solving the problems in chart interpretation tasks.​
2. In the chart-to-code task, the model's performance has obvious advantages compared with open-source models, demonstrating the effectiveness of some designs in the paper.​
3. Sufficient experiments are conducted, including comparative experiments, ablation studies, and qualitative experiments, which help to verify the related hypotheses and results.​
4. The preliminary study on the performance of SFT has certain reference value within the scope of this task, and it is natural to introduce RL based on this experiment, which ensures the logical continuity of the research.

**Weaknesses:**

1. On line 52, the description of the significance of the chart-to-code task is not convincing enough. The authors state that this task is "complex" but fail to highlight its strong "significance". Although it is based on previous research, it may arouse readers' questions about its application value (e.g., help MLLMs better understand charts or assist researchers). In addition, the connection between this part and the previous paragraph is not smooth.​
2. In the preliminary study (Figure 1), the SFT plateau and RL performance gain are shown, but the study on the change of RL performance with the increase of data size is missing. Since SFT will converge when the training data increases to a certain extent, according to the authors' motivation, RL should converge later, but this comparison is lacking. Although I noticed Figures 3 and 4, they only show the change curves of reward. The scale law of specific training data on performance is important and can emphasize the motivation of this paper.​
3. For methods, the main contributions of this paper lie in data synthesis and reward design. The GRPO used is in a standard form (lines 259-269). This not only reduces the innovation of the paper in terms of contributions but also makes the excessive space spent on introducing non-original content unnecessary in writing.​
4. The experimental results support the point 3 that the contribution of synthetic data is significant, which should be the core of this paper. However, the authors do not produce detailed discussion and analysis on data synthesis, usage, scaling law, etc.​
5. In Table 4, the effects of Textual and Textual+Visual are close, but there is no in-depth discussion on this in the paper, which weakens the rationality and effectiveness of the method design.​
6. Table 5 has a similar problem. The results of each variant are close (~1%), but the paper only lists the results without in-depth analysis.​

In general, **the research motivation of this paper needs to be strengthened, the experiments supporting the motivation need to be more completely supplemented, the focus of the designed method is biased, and although the experiments are sufficient, the support and analysis for the method design and problem solving are insufficient.**

**Questions:**

Tthere are some questions and minor issues do not affect my overall assessment:​
1. For Figure C, is there any reference for the selection of these scores?​
2. Regarding citations, the authors have cited a large number of arXiv papers. In fact, many of these papers have been published in conferences, so attention should be paid to the correctness of citations.

---

> ### Author Response · Authors · 2025-11-23
> **Reply (Part 1 / 2)**
>
> Thank you very much for your valuable comments and suggestions. Below, we address each of them in detail.
>
> >W1: Unconvincing and unsmooth descriptions.
>
> Thanks for your valuable comments. In the revised paper, we have emphasized the practical significance and application value of the chart-to-code task, such as its potential to help multimodal large language models better understand charts and to assist researchers in data analysis. We have also improved the coherence of the text to ensure a smoother connection between the adjacent paragraphs.
>
>
> >W2: Missing the study on the change of RL performance with the increase of data size.
>
> Thanks for your suggestion. We have added the performance curves of the two-stage RL training with the increase of data size in Figure 4 of the revised paper. From Figure 4, we can see that:
>
> - The first-stage RL training breaks through the SFT plateau, achieving a significant performance improvement (approximately 3.5%), and converges when reaching a maximum of 22k training data.
>
> - The second-stage RL training involves visual rewards and further improves performance (approximately 1.5%), and almost converges when reaching a maximum of 11k training data.
>
>
> >W3: Non-original content is unnecessary in writing.
>
> Thanks for your comments. We have removed the introduction to GRPO in the "Methods" section and focused on the description of the original contributions in the revised paper.
>
>
> >W4: Detailed discussion and analysis on data synthesis, usage, scaling law, etc.​
>
> Thanks for your insightful suggestions. We have discussed and analyzed the contributions of synthetic data in detail and included them in the revised paper. Below, we discuss the advantages of our synthetic data from the following perspectives.
>
> **Data synthesis:** We detail our data synthesis process and highlight the unique characteristics that differentiate it from existing data. Our data synthesis method offers three main advantages over existing data: a completely realistic data source, more challenging multi-chart content, and richer visual diversity enabled by more API types.
>
> | Method      | Realistic data source | Multi-chart | Chart types | Data samples | API types |
> |-------------|----------------------|-------------|-------------|--------------|-----------|
> | ChartLlama  | no                   | no          | 10          | 11k          | 83        |
> | ChartMoE    | partly               | no          | <20         | 800k         | -         |
> | Chart2Code  | no                   | yes         | 15          | 3k           | 168       |
> | ChartCoder  | no                   | no          | 27          | 115k         | 187       |
> | Ours        | yes                  | yes         | 24          | 3M           | 1,555     |
>
>
> **Data usage:** In Table 5 of the revised paper, we have added detailed data usage for RL, and illustrated the SFT data usage in Figure 1.
>
> **Data effect:** To highlight the value of our synthetic data, we compare the effects of using our synthetic data and open-source data at both the SFT and RL stages.
>
> | Stage | Dataset     | ChartMimic Low-level | High-level |
> |-------|-------------|-----------|------------|
> | SFT   | ChartCoder  | 68.5      | 68.0       |
> |       | Ours        | 73.0      | 77.6       |
> | RL    | ChartCoder  | 76.5      | 81.2       |
> |       | Ours        | 78.6      | 83.8       |
>
> **Data scaling law:** We have presented the scaling law of SFT data in Figure 1 and supplemented the scaling law of RL data in the revised Figure 4.

---

> ### Author Response · Authors · 2025-11-23
> **Reply (Part 2 / 2)**
>
> >W5&W6: The close effects of Textual and Textual+Visual in Tables 4&5 weaken the rationality of method design.
>
> Thanks for your valuable feedback. Although the model performance in Tables 4 and 5 are close, their differences in training resource consumption are significant, which motivates the model design of two-stage RL. Specifically, the model performance of visual reward is superior, but it requires higher training resource consumption due to the substantial time needed for image rendering and model evaluation. To this end, we propose a two-stage RL strategy to trade off performance and consumption: the first stage leverages the low-cost textual reward to improve performance, while the second stage uses a compressed number of visual reward samples to further enhance performance.
>
> In the revised paper, we have provided **new ablation results in Table 5**, discussing the trade-off between the model performance and training resource consumption. The revised ablation study offers the following insights that explain our model design:
> - **Visual reward are more effective:** training with 22K samples with textual or visual reward both provide significant performance improvements, with visual rewards being more effective, especially in high-level evaluation of detailed code generation.
>
> - **Resource consumption is worth considering:** the resource consumption of visual rewards is substantially higher, thus balancing performance and consumption is a crucial issue.
>
> - **Two-stage RL takes a step forward the trade-off:** the first stage uses low-cost textual rewards and the second stage involves high-cost visual rewards with limited samples, balancing efficiency and effectiveness.
>
> Based on these findings, we empirically report the two-stage training strategy using 50% of the training samples in our experiments, as it offers a better trade-off between performance and consumption efficiency.
>
> **Table: Comparison of single-stage and two-stage RL training with different reward strategies.**
>
> | Rewards        | Samples      | ChartMimic Exec | Low-Level | High-Level | Training Cost |
> |----------------|------------|-----------------|-----------|------------|---------------|
> | -              | -          | 93.17           | 73.02     | 77.61      | -             |
> | (T, -)         | (22k, -)   | 97.00           | 78.06     | 82.69      | 1.0           |
> | (V, -)         | (22k, -)   | 97.37           | 79.13     | 84.01      | 5.6           |
> | (T, T + V)     | (22k, 11k) | 97.50           | 79.50     | 83.81      | 3.8           |
> | (T, T + V)     | (22k, 5.5k)| 96.50           | 78.62     | 83.83      | 2.4           |
>
> *Note: “T” and “V” represent textual reward and visual reward, respectively. Training cost is measured in GPU hours. The training cost of each configuration is a multiple of the cost in the second row.*
>
>
> >Q1: For Figure C, is there any reference for the selection of these scores?
>
> We refer to the model evaluation prompt from ChartMimic [1] to set the score ranges for different chart elements.
>
> [1] Yang C, Shi C, Liu Y, et al. ChartMimic: Evaluating LMM's Cross-Modal Reasoning Capability via Chart-to-Code Generation[C]. The Thirteenth International Conference on Learning Representations, 2024.
>
>
> >Q2: Many cited papers are arXiv preprints that have been officially published, so the accuracy of citations should be ensured.
>
> Thank you for pointing this out. We have updated the citations in the revised paper to ensure their accuracy.

---

> > ### Comment · Reviewer_npBT · 2025-11-24
> >
> > I think the authors address my main concern. Therefore, I raise my score to 6. I have some suggestions that the authors may refer to improve the presentation.
> >
> > 1. Figure 4 shows the performance curve of RL with increasing sample number. It should be beneficial to put the SFT curve here to show comparison.
> > 2. Table 5 shows the data and computation efficiency of visual + text reward, which looks reasonable. A confusing point is "The training cost of each configuration is a multiple of the cost in the second row". I think the authors should specify the setting and metric.

---

> > > ### Author Response · Authors · 2025-11-25
> > > **Response to Follow-up Questions**
> > >
> > > Thank you very much for your positive feedback!
> > >
> > > >Q1: It would be beneficial to include the SFT curve in Figure 4 for comparison.
> > >
> > > Thank you for your valuable suggestion. We have updated Figure 4 to include the SFT curve alongside the two-stage RL curve, providing a more direct and intuitive comparison of their performance.
> > >
> > >
> > > >Q2: The description of training cost in Table 5 is confusing and lacks specific metric definitions.
> > >
> > > Thank you for pointing this out. We have replaced the confusing description of "training cost multiples" with the more specific metric of "GPU hours." We further clarify that all experiments were conducted on identical GPUs, ensuring that the reported costs are strictly comparable.
> > >
> > > **Table: Comparison of single-stage and two-stage RL training with different reward strategies.**
> > >
> > > | Rewards        | Samples      | ChartMimic Exec | Low-Level | High-Level | GPU Hours |
> > > |----------------|------------|-----------------|-----------|------------|---------------|
> > > | -              | -          | 93.17           | 73.02     | 77.61      | -             |
> > > | (T, -)         | (22k, -)   | 97.00           | 78.06     | 82.69      | 240           |
> > > | (V, -)         | (22k, -)   | 97.37           | 79.13     | 84.01      | 1344           |
> > > | (T, T + V)     | (22k, 11k) | 97.50           | 79.50     | 83.81      | 912           |
> > > | (T, T + V)     | (22k, 5.5k)| 96.50           | 78.62     | 83.83      | 576           |
> > >
> > > *Note: “T” and “V” represent textual reward and visual reward, respectively. All training is conducted on H800 GPUs.*
> > >
> > >
> > > Thank you again for your valuable suggestions.

---

### Official Review · Reviewer_iWJj · 2025-10-20

**Soundness:** 3
**Presentation:** 3
**Contribution:** 2
**Rating:** 6
**Confidence:** 4

**Summary:**

This paper targets the task of chart-to-code generation, where the model produce excuteable plotting code given a chart image.

The authors observe a performance plateau with supervised fine-tuning (SFT), To overcome this, they propose Multimodal Structured RL (MSRL), a GRPO based reinforcement learning approach with multi-granularity rewards. Training follows a two-stage curriculum: RL on textual feedback, then on combined textual+visual feedback.

The paper claims state-of-the-art results on the ChartMimic and ReachQA benchmarks, with a ~6–10% gain in high-level accuracy over SFT alone.

**Strengths:**

* Clear diagnosis of the SFT plateau. The scaling curve explicitly plateaus after ~2M SFT examples, and shows visual RL breaks the curve.
* Large curated and balanced chart‑code corpus from real‑world arXiv is great contribution to the community.
* MSRL (7B) outperforms ChartCoder and is competitive with GPT‑4o on component scores such as layout and text (Tables 1–2, p. 6), with qualitative cases showing better fidelity and execution reliability than open and proprietary models

**Weaknesses:**

* The visual reward (and RL data filtering) rely on a single MLLM judge (Qwen2.5‑VL‑72B). Without cross‑judge verification or human studies, there is a very risk of reward hacking or bias toward that evaluator’s preferences.
* Possible double‑counting of execution success in stage 2, in Sec 3.3 R = w_t*R_text + w_v*R_vis + W_e*R_exec, R_text also contains R_exec
* Compute-normalized ablation for two-stage vs single-stage: It looks possible that the two-stage curriculum outperforms single-stage primarily because it uses a larger update budget (trained on more steps).
* Several relevant papers are not cited :
    - [1] **Rendering-Aware Reinforcement Learning for Vector Graphics Generation**, Neurips 2025.
    - [2] **MatPlotAgent: Method and Evaluation for LLM-Based Agentic Scientific Data Visualization**, ACL Findings 2024.
    - [3] **A Survey on LLM-as-a-Judge, arXiv:2411.15594**

**Questions:**

* Did you try cross-judge or human judge is done to visual rewards, (e.g., swapping the 72B judge with a different model or use gpt-4o or human to cross-judge qwen2.5-vl-72B, especially at late stage RL, where reward hacking is most possible?
* What's the reason for RL with visual rewards for only 100 steps where the rewards are still gaining fast, is the limit stability, compute, or hacking?
* [weakness #3] Could you disentangle curriculum effects from training budget by reporting a compute-matched comparison?

---

> ### Author Response · Authors · 2025-11-23
> **Reply (Part 1 / 2)**
>
> Thank you very much for your valuable comments and suggestions. Below, we address each of them in detail.
>
> >W1&Q1: Cross-judge or human judge for visual rewards (and RL data filtering).
>
> Thanks for your valuable comments. To verify that the improvement in performance results from effective visual reward design rather than overfitting to the preferences of the VLM judger, we conduct an ablation experiment to observe the distribution of sample scores when the visual evaluator changes. Specifically, we randomly sample 200 instances from the RL data and use the MSRL-SFT model to perform chart-to-code prediction, rendering the resulting Python code into images. Following the ChartMimic scoring guidelines, human ratings of rendered samples serve as the gold scores. Next, we use GPT-4.1, Qwen2.5-VL-72B, Qwen2.5-VL-7B, and InternVL3-38B to score the images. Additionally, we compute the visual similarity between the ground-truth and rendered images using DINOv2-L. We calculate the Pearson correlation coefficients between the scores from these models and the human scores, as shown in the table below. We also plot the distributions of model scores versus human scores, which are provided in Figure A of the revised paper. The results indicate that GPT-4.1 achieves the highest correlation with human judgment and has the most similar score distribution, followed by Qwen2.5-VL-72B, while Qwen2.5-VL-7B performs the worst. Notably, when Qwen2.5-VL-7B is used as the judger, the performance improvement over the SFT model is negligible, demonstrating that model performance is adversely affected when evaluator bias exists. Considering both performance and budget, we select Qwen2.5-VL-72B as the visual evaluator, as it fairly reflects model capability and prevents reward hacking.
>
> **Table: The Pearson correlation coefficients between model scores and human scores.**
>
> | Model               | Pearson Correlation Coefficient |
> |---------------------|-------------------------------|
> | GPT-4.1             | 0.87                          |
> | Qwen2.5-VL-72B      | 0.86                          |
> | Qwen2.5-VL-7B       | 0.51                          |
> | InternVL3-38B       | 0.74                          |
> | DINOv2-L            | 0.71                          |
>
> To address potential MLLM bias in RL data filtering, we manually inspect 100 samples that are assigned a score of 0 by the model, and find over 90% consistency with human judgment. Additionally, we balance the chart type distribution in the final RL dataset to avoid the model preferentially retaining certain chart types. We have added these details to the revised paper.
>
>
> >W2: Possible double‑counting of execution success in stage 2.
>
> Thank you for pointing out this typo. The initially incorrect description pointed to the possible double‑counting of execution success, whereas in fact, R_text does not include R_exec. In the revised paper, we have corrected this typo.
>
>
> >W3&Q3: Compute-normalized ablation for two-stage vs single-stage.
>
> Thank you for your valuable suggestion. We have conducted a compute-normalized ablation to disentangle two-stage curriculum effects from training budget. The results show that, in single-stage RL training, visual rewards yield greater performance improvements compared to textual rewards, but lead to significantly higher training costs. Using both textual and visual rewards in a single stage is computationally expensive, primarily due to the substantial time needed for image rendering and model evaluation. To address this, we propose a two-stage training strategy. The first stage utilizes low-cost textual rewards, while the second stage applies a limited number of high-cost visual rewards. This approach achieves comparable performance at a substantially lower computational cost by effectively combining textual and visual rewards.
>
> **Table: Comparison of single-stage and two-stage RL training with different reward strategies.**
>
> | Rewards        | Samples    | ChartMimic Exec | Low-Level | High-Level | Training Cost |
> |----------------|------------|-----------------|-----------|------------|---------------|
> | -              | -          | 93.17           | 73.02     | 77.61      | -             |
> | (T, -)         | (22k, -)   | 97.00           | 78.06     | 82.69      | 1.0           |
> | (V, -)         | (22k, -)   | 97.37           | 79.13     | 84.01      | 5.6           |
> | (T + V, -)     | (22k, -)   | -               | -         | -          | >5.6          |
> | (T, T + V)     | (22k, 11k) | 97.50           | 79.50     | 83.81      | 3.8           |
> | (T, T + V)     | (22k, 5.5k)| 96.50           | 78.62     | 83.83      | 2.4           |
>
> *Note: “T” and “V” represent textual reward and visual reward, respectively. Training cost is measured in GPU hours. The training cost of each configuration is a multiple of the cost in the second row.*

---

> ### Author Response · Authors · 2025-11-23
> **Reply (Part 2 / 2)**
>
> >Q2: What's the reason for RL with visual rewards for only 100 steps, where the rewards are still gaining fast, is the limit stability, computation, or hacking?
>
> Thanks for your comments. Due to computational constraints, we previously restricted RL training with visual rewards to only 100 steps (i.e., 11k training samples), which is why the reward function did not appear to converge. Based on your suggestion, we continue training the RL process with visual rewards up to 200 steps (22k training samples). The model gradually converges between 100 and 200 steps, achieving higher performance as shown in the table above.
>
>
> >W4: Citing relevant key papers.
>
> Thanks for pointing out these relevant papers, and we have included these papers in the revised paper.

---

> ### Author Response · Authors · 2025-11-28
> **Gentle Reminder: Feedback on Response**
>
> Dear reviewer iWJj：
>
> We are truly grateful for your constructive comments on our submission.
>
> In response to your review, we have supplemented experiments to clarify and answer each of your concerns. Your suggestions are of great significance for improving our work. We would be very grateful and look forward to receiving your feedback on our responses during the discussion.
>
> Thank you for your time and understanding.
>
> Sincerely,
>
> Authors

---

### Official Review · Reviewer_vxZ1 · 2025-10-31

**Soundness:** 3
**Presentation:** 3
**Contribution:** 2
**Rating:** 6
**Confidence:** 4

**Summary:**

The paper studies why SFT saturates on chart-to-code and proposes a two stage RL recipe with multi-granularity rewards (textual + visual). The authors build a large chart-code corpus to show SFT plateaus beyond 2M examples, and then use GRPO with (a) execution + rule-based textual checks and (b) a render and compare visual reward scored by a strong MLLM. On ChartMimic and ReachQA, MSRL (7B) establishes new SOTA among open-source models and approaches GPT-4o.

**Strengths:**

1. Clear empirical story about the SFT ceiling. authors isolate SFT scaling curve and convincingly shows a plateau beyond 2M samples before introducing RL, this sort of strengthens the causal claim that RL brings the next jump, not just under-tuned SFT.
2. The textual reward normalizes code and scores specific fields (data, type, layout, titles/labels, exec) while visual reward compares rendered images against ground truth via an MLLM.. the two-stage schedule is sensible imo and empirically validated.
3. Strong results: MSRL-7B beats open-source baselines and seems competitive with GPT-4o on both datasets.. detailed low-level metrics suggest real gains, not just execution hacks.
4. Repruducible details specified

**Weaknesses:**

- Clearly the visual reward depends on Qwen2.5-VL as judge. If the policy aligns to judge's biases or defects, improvements could reflect evaluator gaming rather than genuine fidelity. Perhaps a judge-swap test (e.g., different MLLM/human) is needed to rule out any judge overfitting.
- The entire pipeline centers on Matplotlib-style code and a fixed rendering toolchain. It is unclear whether the learned behaviors transfer to other plotting libraries (Seaborn/plotly or vega) or different runtimes
- Real-world tables are scraped from arxiv, but the paper doesn’t quantify overlap with benchmarks. If time allows, data contamination studies are a must these days.
- Also, some light on whether the gains are from the reward design vs. the specific optimizer (grpo vs dpo) would be great.
- [Optional but important] The abstract is quite long, and the third paragraph in the introduction (starting around line 75) is similarly lengthy. Try to simplify where possible. Avoid using multiple wrap figures unless absolutely necessary.. if you can, place Table 3 and Table 4 side by side. Also the figure and table captions are too brief and not self-contained. Figure 5 is impossible to read at first glance. It would be good to do a dedicated pass focused on improving writing clarity and presentation quality.

**Questions:**

- How do results change when visual reward/evaluator is swapped (e.g., InternVL (latest), Gemini, GPT-4.1) and on a small human-scored subset?
- What exact de-duplication and topical overlap checks were done to ensure arXiv-derived tables don’t appear in or unduly resemble test items in ChartMimic/ReachQA
- curious to see if MSRL trained on Matplotlib code generalizes to other plotting libraries? Perhaps a notable few?

- Why GRPO? Is it the first thing you tried and worked decently?


Lastly, some relevant key papers on visual code generation that may have been missed in citations:
[1] Rodrigez et al, Generating Scalable Vector Graphics Code From Images And Text. https://arxiv.org/abs/2312.11556
[2] Xia et al, StructChart: Perception, Structuring, Reasoning for Visual Chart Understanding.
[3] Rodriguez et al, BigDocs: An Open Dataset for Training Multimodal Models on Document and Code Tasks. https://arxiv.org/abs/2412.04626
[4] Awal et al. WebMMU: A Benchmark for Multimodal Multilingual Website Understanding and Code Generation https://arxiv.org/abs/2508.16763

Happy to relook at the scores if authors consider some of the points raised.

---

> ### Author Response · Authors · 2025-11-23
> **Reply (Part 1 / 2)**
>
> Thank you very much for your valuable comments and suggestions. Below, we address each of them in detail.
>
> >W1&Q1: Judge-swap test of visual reward evaluator on a small human-scored subset.
>
> Thanks for your suggestion. We perform a judge-swap test of the visual reward evaluator to observe the distribution of sample scores. Specifically, we randomly sample 200 instances from the RL data and use the MSRL-SFT model to perform chart-to-code prediction, rendering the resulting Python code into images. Following the ChartMimic scoring guidelines, human ratings of rendered samples serve as the gold scores. Next, we use GPT-4.1, Qwen2.5-VL-72B, Qwen2.5-VL-7B, and InternVL3-38B to score the images. Additionally, we compute the visual similarity between the ground-truth and rendered images using DINOv2-L. We calculate the Pearson correlation coefficients between the scores from these models and the human scores, as shown in the table below. We also plot the distributions of model scores versus human scores, which are provided in Figure A of the revised paper. The results indicate that GPT-4.1 achieves the highest correlation with human judgment and has the most similar score distribution, followed by Qwen2.5-VL-72B, while Qwen2.5-VL-7B performs the worst. Considering both performance and budget, we select Qwen2.5-VL-72B as the visual evaluator, as it fairly reflects model capability and prevents reward hacking.
>
> **Table: The Pearson correlation coefficients between model scores and human scores.**
>
> | Model               | Pearson Correlation Coefficient |
> |---------------------|-------------------------------|
> | GPT-4.1             | 0.87                          |
> | Qwen2.5-VL-72B      | 0.86                          |
> | Qwen2.5-VL-7B       | 0.51                          |
> | InternVL3-38B       | 0.74                          |
> | DINOv2-L            | 0.71                          |
>
>
> >W2&Q3: Concerns about generalization to other plotting libraries.
>
> Thanks for your comments. To evaluate whether MSRL can transfer its capabilities to other plotting libraries, we collect 107 Seaborn and 150 Plotly images from their official websites. Since we only use Matplotlib-style code to generate charts, the new test data are out-of-domain (OOD). We perform inference with both Qwen2.5-VL-7B and MSRL on these two OOD test sets, calculating execution rates and high-level scores according to the ChartMimic protocol. As shown in the table below, the results demonstrate that MSRL exhibits generalization to unseen Seaborn- and Plotly-style images.
>
> | Model           | Seaborn Exec.Rate | Seaborn High-Level | Plotly Exec.Rate | Plotly High-Level |
> |-----------------|-------------------|--------------------|------------------|-------------------|
> | Qwen2.5-VL-7B   | 69.2              | 25.7               | 62.7             | 22.6              |
> | MSRL            | **85.1**          | **30.5**           | **90.0**         | **35.9**          |
>
>
> >W3&Q2: De-duplication and topical overlap checks over benchmarks.
>
> Thanks for your comments. There is no data leakage. As stated in the ChartMimic paper, they filter arXiv papers published after February 2024 as the source for chart data. To ensure complete separation, we use only papers published in 2023 and earlier as our data source. For ReachQA, the data is generated by LLMs based on the given theme and is not sourced from real-world Arxiv data. We can provide links to all papers used for our data synthesis if required.

---

> ### Author Response · Authors · 2025-11-23
> **Reply (Part 2 / 2)**
>
> >W4&Q4: GRPO vs DPO.
>
> Thanks for your comments. Intuitively, GRPO is naturally more suitable for chart-to-code tasks than DPO. The degree to which code restores a chart can be quantitatively measured by a set of well-defined rules, making it ideal for designing the reward function in GRPO. In contrast, DPO learns from preference pairs, and it is challenging to clearly define what constitutes "bad" code in chart-to-code tasks, which can lead to ambiguity in the model’s optimization direction. To ensure a thorough evaluation, we conduct an ablation study comparing GRPO and DPO. Since DPO requires the construction of positive and negative sample pairs, we follow the chart2code [1] and randomly perturb 2-3 attributes of the chart code to generate negative samples using GPT-4.1, resulting in an RL dataset of comparable size to GRPO for DPO training on the SFT model. As shown in the table below, the performance improvement of DPO is significantly less than that of GRPO. This indicates both the superiority of the GRPO algorithm and the effectiveness of our multimodal structured reward design and two-stage curriculum learning.
>
> | RL Optimizer | ChartMimic Low-Level | High-Level |
> |--------------|----------------------|-----------------------|
> | -            | 73.0                 | 77.6                  |
> | DPO          | 74.0                 | 78.9                  |
> | GRPO         | **78.6**             | **83.8**              |
>
> [1] Zhang Z, Cao Y, Liao L. Enhancing chart-to-code generation in multimodal large language models via iterative dual preference learning[J]. arXiv preprint arXiv:2504.02906, 2025.
>
>
> >W5: Improving writing clarity and presentation quality.
>
> Thank you for your valuable suggestions. In the revised paper, we have simplified the abstract and introduction. We have removed the use of wrap figures. We have refined the captions of all figures and tables that were previously too brief. For Figure 5, we have clarified in the caption that the first column shows the ground-truth images, while the remaining four columns display images rendered from Python code generated by different models, in order to improve readability.
>
>
> >Q5: Citing relevant key papers.
>
> Thanks for pointing out these related works. We have included all these important works in the revised paper.

---

> ### Author Response · Authors · 2025-11-28
> **Gentle Reminder: Feedback on Response**
>
> Dear reviewer vxZ1：
>
> We are truly grateful for your constructive comments on our submission.
>
> In response to your review, we have supplemented experiments to clarify and answer each of your concerns.
> Your suggestions are of great significance for improving our work. We would be very grateful and look forward to receiving your feedback on our responses during the discussion.
>
> Thank you for your time and understanding.
>
> Sincerely,
>
> Authors

---

### Official Review · Reviewer_pbDs · 2025-11-01

**Soundness:** 2
**Presentation:** 2
**Contribution:** 2
**Rating:** 2
**Confidence:** 4

**Summary:**

This paper introduced Multimodal Structured Reinforcement Learning (MSRL) for the chart-to-code task. The authors first analyze the scaling of the supervised finetuning (SFT) approach and show that the performance plateaus beyond certain data/compute scale. After that, they propose an RL-based solution that breaks this plateau and shows further improvements in performance. The authors propose two sets of rewards: textual and visual. The textual reward focuses on aspects in the generated code while the visual rewards measures the similarity between the reference and rendered chart (from the generated code). To create data for SFT and RL, the authors propose a data generation pipeline that first sources tables from arXiv papers. Then, an LLM uses these tables along with code examples to generate code for new chart images. After that, the codes are rendered into chart images which are also further filtered using GPT4o to keep only high fidelity charts.  The authors evaluate their model on two benchmarks: ChartMimic and ReachQA, and show nice performance improvements over existing methods.

**Strengths:**

* The proposed RL approach shows performance improvements on the chart-to-code task that can’t be achieved by scaling the SFT data/training on its own. These experiments are quite interesting and could have valuable insights.


* The proposed model achieves the state-of-the-art results on two chart-to-code benchmarks: ChartMimic and ReachQA.  The authors have provided detailed ablation experiments to show the benefits from each of their proposed techniques/reward functions.

**Weaknesses:**

* Limited Visual diversity: using a set of example codes and forcing the code to follow a specific structure may significantly limit the visual diversity of the dataset. There’s also no analysis to support the claim of visual diversity compared to existing datasets/approaches.


* The evaluation is only limited to two benchmarks: ChartMimic and ReachQA. Furthermore, it’s limited to the niche chart-to-code task. It would strengthen the contribution of the paper if the RL approach can be expanded to other popular chart tasks (e.g., QA) to truly verify the “Breaking the SF Plateau” claim in the title and the paper.


* The visual evaluator in the RL approach is quite massive, 72B params. I am not sure why the authors didn’t simply use a small CNN/ViT and just measure the visual similarity between the extracted features.


* In the Intro, the authors claim that DPO does not generalize well but they haven’t conducted any experiment to prove that GRPO generalizes better than DPO in this task using the same dataset.


* Potential Data Leakage: The ChartMimic benchmark was constructed by scaping data from arXiv and the proposed dataset here also starts by scraping Tables from arXiv. I am concerned that the performance improvements could be tied to data leakage.

**Questions:**

see weaknesses above

---

> ### Author Response · Authors · 2025-11-23
> **Reply (Part 1 / 2)**
>
> Thank you very much for your valuable comments and suggestions. Below, we address each of them in detail.
>
> >W1: The dataset may lack visual diversity due to the use of example codes and enforced code structure, and there is no comparative analysis with existing datasets.
>
> Thanks for your insightful comments. We use example codes to enhance generation quality through in-context demonstration. Here, we provide a comparative analysis with existing datasets to prove the visual diversity of our dataset. As shown in the table below, the broad coverage of API types in our synthetic data results in higher code complexity, thereby promoting greater visual diversity. Notably, our data is entirely derived from realistic data sources, features more challenging multi-chart content, and contains the largest number of samples among datasets with a comparable number of chart types.
>
> | Method      | Realistic data source | Multi-chart | Chart types | Data samples | API types |
> |-------------|----------------------|-------------|-------------|--------------|-----------|
> | ChartLlama  | no                   | no          | 10          | 11k          | 83        |
> | ChartMoE    | partly               | no          | <20         | 800k         | -         |
> | Chart2Code  | no                   | yes         | 15          | 3k           | 168       |
> | ChartCoder  | no                   | no          | 27          | 115k         | 187       |
> | Ours        | yes                  | yes         | 24          | 3M           | 1,555     |
>
>
> >W2: Extend the RL approach to other charting tasks such as QA to verify the claim of "Breaking the SF Plateau".
>
> Thanks for your suggestion. We validate the generalization of our RL approach on the ChartQA dataset. Qwen2.5-VL-7B achieves 87.3 on the ChartQA test set, but its performance decreases to 86.2 after SFT training on the ChartQA training set, demonstrating the SFT bottleneck. Subsequently, when we apply RL training with text-based reward using the same ChartQA training set, the test performance improves to 89.2. These results further support the claim of "Breaking the SFT Plateau" stated in our paper. However, as this experiment is not closely aligned with the main theme of the paper (i.e., Chart-to-Code Generation), we do not include it in the submission.
>
> | Model               | ChartQA      |
> |---------------------|--------------|
> | Qwen2.5-VL-7B       | 87.3         |
> | Qwen2.5-VL-7B-SFT   | 86.2         |
> | Qwen2.5-VL-7B-RL    | 89.2         |
>
>
> >W3: The RL approach uses a large visual evaluator (72B parameters) instead of a smaller model like CNN or ViT for measuring visual similarity.
>
> Thanks for your suggestions. We utilize Qwen2.5-VL-72B as the visual evaluator due to its superior model performance and better pearson correlation coefficient with human judgment. To demonstrate this, we have added two more experiments in the revised paper, including evaluating model performance and reward correlation for different visual evaluator.
>
> - **Model performance:** We compare two small vision models, InternViT and DINOv2-L, as evaluators for visual rewards, calculating the visual similarity between ground-truth and generated image features. As shown in the table below, these vision models yield improved performance. However, a substantial gap remains compared to models employing VLMs as visual evaluators. This indicates that vision models trained on image-text matching assess similarity at a coarse level and do not adequately capture fine-grained differences in chart elements.
>
> - **Reward correlation:** We compare the Qwen2.5-VL-72B with a small model DINOv2-L, then calculate the pearson correlation coefficient between their measured visual similarity scores and human scores on a small curated test set. From the results below, the smaller vision models such as DINOv2-L exhibit a lower correlation coefficient in score distributions compared to Qwen2.5-VL-72B, indicating worse performence in measuring visual similarity for our proposed RL approach.
>
> The above results indicate that the selection of Qwen2.5-VL-72B is reasonable.
>
>
> | Visual Evaluator     | Pearson Correlation Coefficient | ChartMimic Low-Level |  High-Level |
> |----------------------|-------------------------------|----------------------|-----------------------|
> | -                    | -                             | 73.0                 | 77.6                  |
> | InternViT            | -                             | 75.1                 | 78.1                  |
> | DINOv2-L             | 0.71                          | 75.5                 | 79.8                  |
> | Qwen2.5-VL-72B       | **0.86**                          | **77.8**             | **81.9**              |

---

> > ### Author Response · Authors · 2025-11-23
> > **Reply (Part 2 / 2)**
> >
> > >W4: The claim that GRPO generalizes better than DPO is not supported by experiments on the same dataset.
> >
> > Thanks for your valuable comments. We conduct additional comparative experiments between GRPO and DPO on the same dataset to assess their generalization performance. Since DPO requires the construction of positive and negative sample pairs, we follow the chart2code [1] and randomly perturb 2-3 attributes of the chart code to generate negative samples using GPT-4.1, resulting in an RL dataset of comparable size to GRPO for DPO training on the SFT model. The experimental results show that the performance improvement of DPO is significantly less than that of GRPO, demonstrating the effectiveness of our multimodal structured reward design in GRPO algorithm. The advantage of GRPO mainly comes from the flexible reward designs tailored to the chart, rather than the limited negative sample design in DPO.
> >
> > | RL Optimizer | ChartMimic Low-Level | High-Level |
> > |--------------|----------------------|-----------------------|
> > | -            | 73.0                 | 77.6                  |
> > | DPO          | 74.0                 | 78.9                  |
> > | GRPO         | **78.6**             | **83.8**              |
> >
> > [1] Zhang Z, Cao Y, Liao L. Enhancing chart-to-code generation in multimodal large language models via iterative dual preference learning[J]. arXiv preprint arXiv:2504.02906, 2025.
> >
> >
> > >W5: There is concern about potential data leakage since both the ChartMimic benchmark and the proposed dataset are constructed by scraping tables from arXiv.
> >
> > Thanks for your comments. There is no data leakage because the data sources are collected from non-overlapping time periods. As stated in the ChartMimic paper, they filter arXiv papers published after February 2024 as the source for chart data. To ensure complete separation, we only use papers published in 2023 and earlier as our data source. We can provide links to all papers used for our data synthesis if required.

---

> ### Comment · Reviewer_pbDs · 2025-11-26
>
> Thank you for the rebuttal which has addressed some of my concerns. I have raised my score accordingly.

---

> > ### Author Response · Authors · 2025-11-27
> > **Reply**
> >
> > Thank you very much for your positive feedback! We are very glad that our reply can address some of your concerns. We attach great importance to your opinion and look forward to resolving your remaining concerns.

---

### Author Response · Authors · 2025-11-23
**List of Revisions**

We sincerely appreciate your thoughtful and constructive feedback. We attach great importance to every comment and suggestion. According to the reviewers' suggestions, we have submitted a revised version of the paper. The main revisions are as follows:

-In **Section 3.1**, we have **quantified the overlap with benchmarks**, to address the data leakage concerns raised by reviewer pbDs and reviewer vxZ1.

-In **Section 3.1 and Section 4.2**, we have added **detailed discussion and analysis on data contribution**, including data synthesis, data usage, data effect and data scaling law, to address the concerns of visual diversity raised by reviewer pbDs and insufficient support raised by reviewer npBT.

-In **Section 4.2**, we have added the study on the **change of RL performance with the increase of data size**, to address the concerns raised by reviewer npBT.

-In **Section 4.3**, we have inclueded **two out-of-domain test datasets**, to address the concerns of insufficient benchmarks raised by reviewer pbDs and generalizability to other plotting libraries raised by reviewer vxZ1.

-In **Section 4.3**, we have discussed the **trade-off between the model performance and training resource consumption**, to address the concerns of rationality of method design raised by reviewers iWJj and reviewer npBT.

-In **Appendix C**, we have conducted **a judge-swap test and reward correlation analysis for visual reward evaluator**, to address the concerns about not using a smaller visual model raised by reviewer pbDs and the concerns of judge overfitting during the RL raised by reviewers vxZ1 and iWJj.

-In **Appendix C**, we have added **the experimental comparison between GRPO and DPO**, to address the concerns of algorithm validity raised by reviewer pbDs and reviewer vxZ1.

-In addition, we have simplified the abstract and introduction, improved persuasiveness and fluency, and adjusted the structure and content of images and tables, to address the concerns of paper writing raised by reviewer vxZ1 and reviewer npBT.

---

### Author Response · Authors · 2025-12-01
**Summary of Revisions and Reviewer Feedback**

Dear AC,

We thank you and the reviewers for the time and care invested in reviewing our submission. Below is a summary of (1) the strengths acknowledged by the reviewers, and (2) how we addressed their concerns in our rebuttal and revised draft.

**Reviewers acknowledged several strengths in our work.** Our method clearly identifies and overcomes the SFT performance plateau in chart-to-code generation. We propose a novel multimodal RL approach and contribute the largest real-world chart-code dataset to date. Our experiments demonstrate strong and reliable improvements over existing methods. All four reviewers acknowledged the novelty and impact of our approach.

**We have made substantial efforts to address reviewers’ concerns.** We quantified data overlap with benchmarks to avoid leakage, added detailed discussions and analysis on data synthesis, usage, and scaling laws, and incorporated out-of-domain test datasets to demonstrate generalizability. We also studied RL performance scaling and the trade-off between model performance and resource consumption. To address concerns about visual evaluator bias and algorithm validity, we added judge-swap tests, reward correlation analysis, and direct comparisons between GRPO and DPO, and improved the clarity and structure of the paper as suggested.

**Reviewer pbDs and Reviewer npBT raised their scores prior to the OpenReview bug incident after our rebuttal and confirmed their main concerns were addressed.** Reviewer vxZ1 said he would reconsider his score if our revision addressed his points. We believe our changes resolved his concerns, and with more time, his score would likely have increased.

We have paid significant effort during the rebuttal period, and we sincerely hope that these additions and clarifications help further demonstrate the value of our work. We once again thank all reviewers and the AC for their time and dedication.

Sincerely,

Authors

---

### Meta-Review · Area_Chair_nJ31 · 2026-01-06

**Summary:**

This paper systematically investigates the performance plateau of supervised fine-tuning (SFT) through large-scale empirical analysis and introduces Multimodal Structured Reinforcement Learning (MSRL) for chart-to-code generation. In addition, the authors construct the largest training corpus to date, comprising approximately 3 million chart–code pairs curated from real-world tables in arXiv papers, effectively addressing the limitations of prior approaches that rely heavily on synthetic datasets. The initial scores were 6, 2, 4, and 6. During the rebuttal phase, the reviewers with lower initial scores (2 and 4) increased their ratings. In the rebuttal, the authors effectively addressed the concerns regarding potential data leakage, added additional evaluations on diverse test datasets, and included experimental comparisons between GRPO and DPO. They also provided more detailed discussion and analysis of data-related factors, including data synthesis, data usage, data effectiveness, and scaling behavior. Overall, the rebuttal resolved most of the reviewers’ concerns, and therefore I recommend acceptance.

**Reviewer Concerns:**

Most of the reviewers’ concerns have been addressed. The authors are encouraged to carefully revise the final version by following the comments from Reviewers vxZ1, iWJj, and npBT, particularly regarding issues related to presentation, clarity, and typos.

**Reviewer Scores:**

The reviewers with initial scores of 2 and 4 already have increased their scores after the rebuttal, while the reviewers who initially assigned scores of 6 would likely have kept their scores unchanged.

---

### Decision · Program_Chairs · 2026-01-26

Accept (Poster)